# Stratospheric carbon isotope fractionation and tropospheric histories of CFC-11, CFC-12 and CFC-113 isotopologues

Max Thomas[1], Johannes C. Laube[1,2], Jan Kaiser[1], Samuel Allin[1], Patricia Martinerie[3],

Robert Mulvaney[4], Anna Ridley[1], Thomas Röckmann[5], William T. Sturges[1], and Emmanuel Witrant[6]

[1]Centre for Ocean and Atmospheric Sciences, School of Environmental Sciences, University of East Anglia, Norwich

[2]Institute of Energy and Climate Research – Stratosphere (IEK-7), Forschungszentrum Jülich GmbHJ, Jülich, Germany

[3]Univ. Grenoble Alpes, CNRS, IRD, Grenoble INP, IGE, 38000 Grenoble, France

[4]British Antarctic Survey, Cambridge, UK

[5]Institute for Marine and Atmospheric Research, Utrecht University, Utrecht, the Netherlands.

[6]Université Grenoble Alpes, CNRS, Grenoble Image Parole Signal Automatique (GIPSA-Lab), Grenoble, France

**Correspondence:** Max Thomas (max.thomas@uea.ac.uk)

**Abstract.** We present novel measurements of the carbon isotope composition of CFC-11 ($CCl_3F$), CFC-12 ($CCl_2F_2$), and CFC-113 ($CF_2ClCFCl_2$), three atmospheric trace gases that are important for both stratospheric ozone depletion and global warming. These measurements were carried out on air samples collected in the stratosphere – the main sink region for these gases – and on air extracted from deep polar firn snow. We quantify, for the first time, the apparent isotopic fractionation, $\epsilon_{app}(^{13}C)$, for these

gases as they are destroyed in the high- and mid-latitude stratosphere: $\epsilon_{app}$(CFC-12, high-lat) $= (-20.2 \pm 4.4)$ ‰ and $\epsilon_{app}$(CFC-113, high-lat) $= (-9.4 \pm 4.4)$ ‰, $\epsilon_{app}$(CFC-12, mid-lat) $= (-30.3 \pm 10.7)$ ‰ , and $\epsilon_{app}$(CFC-113, mid-lat) $= (-34.4 \pm 9.8)$ ‰. Our CFC-11 measurements were not sufficient to calculate $\epsilon_{app}$(CFC-11) so we instead used previously reported photolytic fractionation for CFC-11 and CFC-12 to scale our $\epsilon_{app}$(CFC-12), resulting in $\epsilon_{app}$(CFC-11, high-lat) $= (-7.8 \pm 1.7)$ ‰ and $\epsilon_{app}$(CFC-11, mid-lat) $= (-11.7 \pm 4.2)$ ‰. Measurements of firn air were used to construct histories of the tropospheric isotopic

composition, $\delta_T(^{13}C)$, for CFC-11 (1950s to 2009), CFC-12 (1950s to 2009), and CFC-113 (1970s to 2009) — with $\delta_T(^{13}C)$ increasing for each gas. We used $\epsilon_{app}$(high-lat), which were derived from more data, and a constant isotopic composition of emissions, $\delta_E(^{13}C)$, to model $\delta_T(^{13}C, CFC-11)$, $\delta_T(^{13}C, CFC-12)$, and $\delta_T(^{13}C, CFC-113)$. For CFC-11 and CFC-12, modelled $\delta_T(^{13}C)$ was consistent with measured $\delta_T(^{13}C)$ for the entire period covered by the measurements, suggesting no dramatic change in $\delta_E(^{13}C, CFC-11)$ or $\delta_E(^{13}C, CFC-12)$ has occurred since the 1950s. For CFC-113, our modelled $\delta_T(^{13}C, CFC-113)$

did not agree with our measurements earlier than 1980. This discrepancy may be indicative of a change in $\delta_E$(13C, CFC-113).

However, this conclusion is based largely on a single sample and only just significant outside the 95 % confidence interval. Therefore more work is needed to independently verify this temporal trend in the global tropospheric $^{13}$C isotopic composition of CFC-113. Our modelling predicts increasing $\delta_T(^{13}C, CFC\text{-}11)$, $\delta_T(^{13}C, CFC\text{-}12)$, and $\delta_T(^{13}C, CFC\text{-}113)$ into the future. We investigated the effect of recently reported new CFC-11 emissions on background $\delta_T(^{13}C, CFC\text{-}11)$ by fixing model emissions after 2012, and comparing $\delta_T(^{13}C, CFC\text{-}11)$ in this scenario to the model base case. The difference in $\delta_T(^{13}C, CFC\text{-}11)$ between these scenarios was 1.4 ‰ in 2050. This difference is smaller than our model uncertainty envelope and would therefore require improved modelling and measurement precision, as well as better quantified isotopic source compositions, to detect.

*Copyright statement.* TEXT

## 1 Introduction

Chlorofluorocarbons (CFCs) have been produced since the 1940s for multiple uses, such as refrigerant gases, aerosol propellants, and in foam blowing. Since their role in ozone depletion was discovered (Molina and Rowland, 1974; Farman et al., 1985), there has been a global effort to phase out the production and use of CFCs, culminating in the essentially complete elimination of their production in 2015 under the Montreal protocol. Long-term monitoring shows that the atmospheric mole fractions of the three most abundant CFCs (CFC-11, -12 and -113) have been declining as a result (Carpenter et al., 2014). However, recent studies highlight the need for continued, careful monitoring of CFCs. Montzka et al. (2018) found evidence for a recently emerged source of atmospheric CFC-11, with subsequent studies tracing these emissions largely to north-east China (Rigby et al., 2019; Adcock et al., 2020). In addition, Adcock et al. (2018) found increasing mole fractions of CFC-113a. Isotopic measurements could provide additional constraints when identifying sources and sinks of CFCs.

CFCs are released to the troposphere by industrial processes and emission from existing banks (Lickley et al., 2020). Once in the troposphere, CFCs are transported to the stratosphere where they are subject to UV photolysis and reaction with $O(^1D)$. Loss by photolysis is dominant, with loss by $O(^1D)$ contributing around 2 %, 6 %, and 6% for CFC-11, -12, and -113, respectively (Burkholder et al., 2013). The balance of these sources and sinks, and the transport processes between them, determines the atmospheric lifetime of a CFC and its tropospheric concentration.

These processes also influence the isotopic signature of CFCs. Breakdown in the stratosphere preferentially destroys light isotopologues, causing a fractionation that leaves the un-photolysed stratospheric CFC pool enriched in heavy isotopes – $^{13}$C and $^{37}$Cl – relative to the troposphere. Such behaviour has been observed for $\delta(^{37}Cl, CFC\text{-}11)$, $\delta(^{37}Cl, CFC\text{-}12)$, and $\delta(^{37}Cl, CFC\text{-}113)$ (Allin et al., 2015; Laube et al., 2010a), and for other gases, such as $N_2O$ (Griffith et al., 2000; Rahn and Wahlen, 1997; Röckmann et al., 2001; Kaiser et al., 2006; Toyoda et al., 2018), $CH_4$ and $H_2$ (Röckmann et al., 2003b; Rahn et al., 2003; Rhee et al., 2006; Röckmann et al., 2011). Heavy isotopologues of CFCs are enriched in the troposphere when this stratospheric pool mixes with the troposphere. There is a good conceptual understanding of isotopic budgets of CFCs, but significant uncertainties remain that hinder the use of isotopic methods to study CFC emissions, sources, and sinks.

One such uncertainty is the degree to which CFCs fractionate in the stratosphere. To date, few studies have been carried out to quantify the apparent isotopic fractionation, $\epsilon_{app}$, in CFCs. $\epsilon_{app}$ relates the change in isotopic signature of a chemical to the degree of destruction observed in the atmosphere using a Rayleigh fractionation model. It is an empirical value that is affected by intrinsic photochemical fractionation, destruction by $O(^1D)$, and transport and mixing (Kaiser et al., 2006).

In the dominant stratospheric sink region, photochemical loss dominates loss from reaction with $O(^1D)$ for CFC-11 and -12 (Minschwaner et al., 2013). Laube et al. (2010b) measured vertical profiles of $\delta(^{37}Cl, CFC-12)$ in stratospheric air from tropical latitudes, calculating $\epsilon_{app}(^{37}Cl, CFC-12) = (-12.1 \pm 1.7)\,‰$. Using similar methodology, Allin et al. (2015) calculated $\epsilon_{app}(^{37}Cl, CFC-12) = (-12.2 \pm 1.6)\,‰$ at mid- and $(-6.8 \pm 0.8)\,‰$ high-latitudes. This decrease in the magnitude of $\epsilon_{app}(^{37}Cl)$ with increasing latitude is qualitatively consistent with observations of $\delta(^{15}N, N_2O)$ and $\delta(^{18}O, N_2O)$ (Kaiser et al., 2006),

though the decrease is larger for CFC-12. Allin et al. (2015) observed no latitude dependence for $\epsilon_{app}(^{37}Cl, CFC-11)$ (mid: $(-2.4 \pm 0.5)\,‰$, high: $(-2.3 \pm 0.4)\,‰$) and $\epsilon_{app}(^{37}Cl, CFC-113)$ (mid: $(-3.5 \pm 1.5)\,‰$, high: $(-3.3 \pm 1.2)\,‰$), though they speculated that some latitude dependence could be obscured by their uncertainties.

For CFCs, the only study of $\epsilon_{app}(^{13}C)$ – from here, $\epsilon_{app}$ – under conditions representative of the stratosphere was the laboratory photolysis experiment of Zuiderweg et al. (2012). Laboratory experiments exclude the effects of atmospheric transport and

mixing, which tend to dilute observed fractionations such that $\epsilon_{app}$ tends to be less than intrinsic photolytic fractionations, $\epsilon_p$, (Kaiser et al., 2006). Zuiderweg et al. (2012) reported $\epsilon_p$ under stratospherically relevant conditions for CFC-11 (($-23.8 \pm 0.9$) ‰ at 203 K to ($-23.0 \pm 1.1$) ‰ at 233 K) and CFC-12 (($-66.2 \pm 3.1$) ‰ at 203 K to ($-55.3 \pm 3.0$) ‰ at 233 K). These values imply greater levels of fractionation for $\delta(^{13}C)$ than for $\delta(^{37}Cl)$ in the stratosphere.

Another uncertainty in our understanding of CFC isotopologues is the isotopic signature of their sources. Allin et al. (2015)

used their measured $\epsilon_{app}$ for CFC-11, CFC-12, and CFC-113 to model a tropospheric history of $\delta(^{37}Cl)$ in these chemicals, following the approach of Röckmann et al. (2003a). Allin et al. (2015) constructed a tropospheric history of the isotopic composition of these chemicals from measurements of tropospheric and firn air — deep, compacted snow containing an archive of tropospheric air going back decades (e.g. Buizert et al., 2012). When a constant isotopic source signature was assumed, the model agreed well with measurements of $\delta(^{37}Cl)$ representative of tropospheric air from around 1970 onwards. Five pre-1970

air samples had $\delta(^{37}Cl)$ values that were inconsistent with the model. However, no clear trend was observable for these five samples and, in addition, the disagreement was not significant to $2\sigma$. The authors concluded that a constant source signature is likely consistent with measured $\delta(^{37}Cl)$ since 1970, and that, with current measurement precisions, it is premature to assign a source change to CFC-11, -12, and -113 in the period before this.

In contrast, Zuiderweg et al. (2013) presented evidence for a past change in $\delta(^{13}C, CFC-12)$. Large depletions, around -40

‰ relative to the present day troposphere, were measured in one deep firn air sample that corresponded to a mean age of around 1965. A significant change in the source signature of CFC-12 is required to explain this observation, and Zuiderweg et al. (2013) suggest that a change in feedstock during CFC production is the most promising explanation. But the results of Zuiderweg et al. (2013) rely heavily on one firn air sample that was potentially biased due to interference from a nearby chromatographic peak (see below and Appendix C). The tropospheric history of $\delta(^{13}C, CFC-12)$ remains uncertain.

**Table 1.** Samples analysed in this study.

| Sample type | Sampling location | Sampling date | Time period covered by samples | Analytical method |
|---|---|---|---|---|
| High latitude stratosphere | Aircraft flights out of Kiruna[†] | December 2011 | December 2011 | A |
| Mid latitude stratosphere | Balloon launched out of Gap[*] | June 1999 | June 1999 | A |
| Firn air | Northern Greenland[+] | 21–30 July 2008 | ≈1997 to 2008[††] | B |
| Firn air | Northern Greenland[§] | 15–24 July 2009 | ≈1955 to 2009[††] | B |

[†] 62 to 72 °N, 2 °W to 24 °E; 9–19 km; Laube et al. (2010b)
[*] 44.4 to 44.8 °N, 3.1 to 6.3 °E; 8–34 km; Kaiser et al. (2006)
[+] 77.45 °N, 51.06 °W; Buizert et al. (2012)
[§] 77.45 °N, 51.06 °W; Zuiderweg et al. (2013)
[††] Mean ages of the age distributions of the firn air samples

We re-measured the firn profile analysed by Zuiderweg et al. (2013) using a method similar to Allin et al. (2015) and Laube et al. (2010a), to better constrain and independently assess the history of $\delta(^{13}C, CFC\text{-}12)$. Together with $\delta(^{13}C, CFC\text{-}12)$, we also measured $\delta(^{13}C, CFC\text{-}11)$ and $\delta(^{13}C, CFC\text{-}113)$ in firn and stratospheric air samples. For the first time, stratospheric measurements were used to calculate $\epsilon_{app}(^{13}C)$ for CFC-11, -12, and -113. We then used these $\epsilon_{app}$ values to quantify the isotope

effect associated with the stratospheric removal in a two box model, using a constant isotopic source signature, and calculated the temporal evolution of tropospheric $\delta(^{13}C)$ for these chemicals since 1937 (CFC-12), 1946 (CFC-11), and 1962 (CFC-113). Model results were compared to the firn measurements to investigate whether changes in isotopic source composition are required to explain the $\delta(^{13}C)$ history of these chemicals.

## 2 Methodology

### 2.1 Sample collection

We present new data from two stratospheric and two tropospheric data sets (Table 1). One stratospheric data set, which we call 'Kiruna', was collected at high-latitudes from high altitude Geophysica flights out of Kiruna, Sweden, using the BONBON-I, BONBON-II, and CLAIRE cryogenic whole air samplers (Laube et al., 2010b). The other, which we call 'Gap', was collected at mid-latitudes from balloons launched from Gap, France, using the whole air sampler of the then Max Planck Institute for

Aeronomy (Kaiser et al., 2006). The firn air samples were collected at NEEM in northern Greenland during field campaigns in 2008 and 2009. Shallow ice cores were drilled, stopping every few meters for air sampling, with the borehole sealed off from ambient air using a bladder (Allin et al., 2015; Schwander et al., 1993).

### 2.2 Sample preparation and analysis

All of the samples were analysed using an established gas chromatography (GC)/mass spectrometry (MS) method that has

produced robust and consistent results for chlorine isotope ratios in natural abundance CFC-11, -12, and -113 (Laube et al.,

2010b; Allin et al., 2015). In short, an Agilent 6890 GC was coupled to a VG/Waters tri-sector mass spectrometer. Air samples were dried by passing them through magnesium perchlorate granules, before being concentrated onto a Hayesep D 80/100 mesh held at -78 $^o$C in a sample loop using a dry-ice/ethanol mixture. Desorption from the Hayesep D was achieved by heating the sample loop to around 95 $^o$C using hot water. A high purity helium stream transferred the sample to a 0.32 mm internal diameter, GS-GasPro (30 m) or KCl-passivated CP-PLOT Al$_2$O$_3$ (50 m) column held at -10 $^o$C. The column was heated at 10 $^o$C min$^{-1}$ to 200 $^o$C to release the chemicals of interest, separated by their retention strength, and pass them to the MS. Every 4$^{th}$ injection was a standard. While the above method was used for every sample analysed, different volumes were trapped for the stratospheric samples (200 ml at 20 $^o$C and 1 bar, method A) and the NEEM 2008/09 firn samples (600 ml at 20 $^o$C and 1 bar, method B). Also, method A used similar instrument settings to Allin et al. (2015). Method B uses the same MS and chromatography, but we increased the detector voltage (from 375 to 400 V), reduced the number of mass fragments measured at any given time, and optimised our source and collector slit parameters for maximum signal. Measurements of $\delta(^{37}$Cl, CFC-11), $\delta(^{37}$Cl, CFC-12), and $\delta(^{37}$Cl, CFC-113) in firn and stratospheric air samples using method A were presented previously by Allin et al. (2015). In this study, we use method A (Allin et al., 2015) and method B, which differers from method A by slight changes to the instrument settings and by looking at different ion fragments (Section 2.3). We further validate our method in Appendix B, where we show that our method produces $\delta(^{13}$C, CFC-12) that are consistent with a GC-IRMS system over wide range of $\delta(^{13}$C, CFC-12).

Our method allows measurements of $\delta(^{37}$Cl) (Allin et al., 2015) and $\delta(^{13}$C) (this study) for CFCs with main isotopologues in the pmol/mol range. A key advantage of our method is that we can make these measurements using only a few hundred millilitres of air, which is important when measuring typical stratospheric and firn air samples where sample volumes are restricted.

## 2.3 Data processing

$\delta(^{13}$C) was calculated using

$$\delta(^{13}\text{C}) = \frac{R_{\text{samp}}(102/101)}{R_{\text{std}}(102/101)} - 1, \tag{1}$$

where $R_{\text{samp}}(102/101)$ and $R_{\text{std}}(102/101)$ are the ratios of the $^{13}$C$^{35}$Cl$_2$F$^+$ ($m/z = 101.9 \approx 102$) to $^{12}$C$^{35}$Cl$_2$F$^+$ ($m/z = 100.9 \approx 101$) ion fragments for the sample and the reference gas, respectively. $R_{\text{std}}$ was taken to be the weighted mean ratio of two preceding and subsequent reference gases. Measured $\delta(^{13}$C, CFC-11) and $\delta(^{13}$C, CFC-12) reflect the total fractionation of each gas whereas $\delta(^{13}$C, CFC-113) only reflects the fractionation on the CCl$_2$F fragment, neglecting fractionation on the CClF$_2$ fragment. The reference used for all measurements was 2005 Northern Hemisphere background air, AAL-071170 — from here referred to as 'AAL'. For the stratospheric samples (method A), $R(102/101)$ was calculated by regressing separate raw intensities for each ion fragment against each other (Laube et al., 2010a; Allin et al., 2015). For the firn samples (method B, using a larger air volume) the intensity of the $m/z101$ fragment saturated the detector and we instead used the $^{12}$C$^{37}$Cl$_2$F$^+$ ($m/z = 104.9 \approx 105$) fragment to calculate $R(102/105)$, again by the regression of the separate raw intensities for each

fragment. To recover $R(102/101)$, we applied a correction to the measured $R(102/105)$ based on the expected $R(105/101)$ using the relation

$$\frac{R_{\text{samp}}(102/101)}{R_{\text{std}}(102/101)} = \frac{R_{\text{samp}}(102/105)}{R_{\text{std}}(102/105)} \cdot \frac{R_{\text{samp}}(105/101)}{R_{\text{std}}(105/101)}. \tag{2}$$

The expected $R(105/101)$ values correspond to $^{12}C^{37}Cl_2F_2/^{12}C^{35}Cl_2F_2$ isotopologue ratios (relative mass difference of 4) and were calculated based on the modelling of diffusive/gravitational fractionation in firn (Section 2.4), assuming a constant isotopic source composition. In this case, $\frac{R_{\text{samp}}(105/101)}{R_{\text{std}}(105/101)} = 1 + 4c$, with $c$ being the correction for a relative mass difference of 1. Substituting Equation 2 into 1 allows us to recover $\delta$ values from measurements of $R(102/105)$. With this treatment we use $^{12}C^{37}Cl_2F_2$ as a standard, assuming $\delta_T(^{37,37}Cl)$ for CFC-11, -12, and -113 is determined dominantly by diffusive and gravitational fractionation in the firn. As a check on our correction, we plot depth profiles of $2c$ and $\delta(^{37}Cl)$ measurements of the NEEM 2009 firn profile presented in Allin et al. (2015) (Figure 1). $2c$ is similar to $\delta(^{37}Cl)$, within the precision of the data, as expected considering Allin et al. (2015) did not observe temporal signals in $\delta(^{37}Cl)$. Given the lack of temporal signals in $\delta(^{37}Cl)$, we do not expect large temporal changes in tropospheric $\delta(^{37,37}Cl)$. The median magnitude of the effect of this correction on our $\delta(^{13}C)$ is 0.8 ‰ (CFC-11), 0.7 ‰ (CFC-12), and 0.5 ‰ (CFC-113). The impact of the correction on $\delta(^{13}C)$ increases with depth in the firn, reaching maximum magnitudes of 4.2 ‰ (CFC-11), 4.3 ‰ (CFC-12), and 0.9 ‰ (CFC-113).

To ascertain the linearity of the response of our analytical system, we performed dilution series for both methods, as described in Appendix A and shown in Figure A1. For method A, the dilution series showed that below a certain threshold (minimum peak area) there are systematic deviations in our measurement methods. Based on these results, a number of stratospheric samples, for which the peak area fell blow this threshold, were excluded. From a total of 38 measurements of each CFC, we rejected 22 (CFC-11), 9 (CFC-12), and 12 (CFC-113) measurements because they fell outside of the linearity limit of our method. For method B, we did not reject any of the 56 measurements performed based on the dilution series. We did, however, exclude the 69.4 m and 71.9 m NEEM 2009 samples (10 measurements) for CFC-113 because, for the corresponding mean ages, there was too little CFC-113 in the atmosphere (Adcock et al., 2018) to reliably determine $\delta(^{13}C)$ values. These 10 measurements are shown in Figure A1 but did not contribute to our analysis.

## 2.4 Modelling firn air transport

Differing masses and diffusivities cause gases, and isotopologues of a given gas, to move through firn at different rates. Here, a model of gas transport in firn air (Witrant et al., 2012) was used to predict both the age distribution for CFC-11, -12, and -113 at each firn sampling depth, and the gravitational and diffusive fractionation of each of these CFCs, using a constant isotopic source composition. The gravitational/diffusive corrections for a relative mass difference of 1 ($c$ in Section 2.3 and Figure 1) range from -1.0 ‰ to 0.2 ‰ (CFC-11); -1.1 ‰ to 0.2 ‰ (CFC-12); and -0.2 ‰ to 0.2 ‰ (CFC-113). In the upper firn, enrichment due to gravitational fractionation gives positive $c$; while in the deeper firn, $c$ is negative as diffusive fractionation overwhelms the gravitational fractionation. Once measurements of firn air are corrected for this fractionation, any change in $\delta(^{13}C)$ is indicative of changes in the tropospheric isotopic composition, $\delta_T(^{13}C)$.

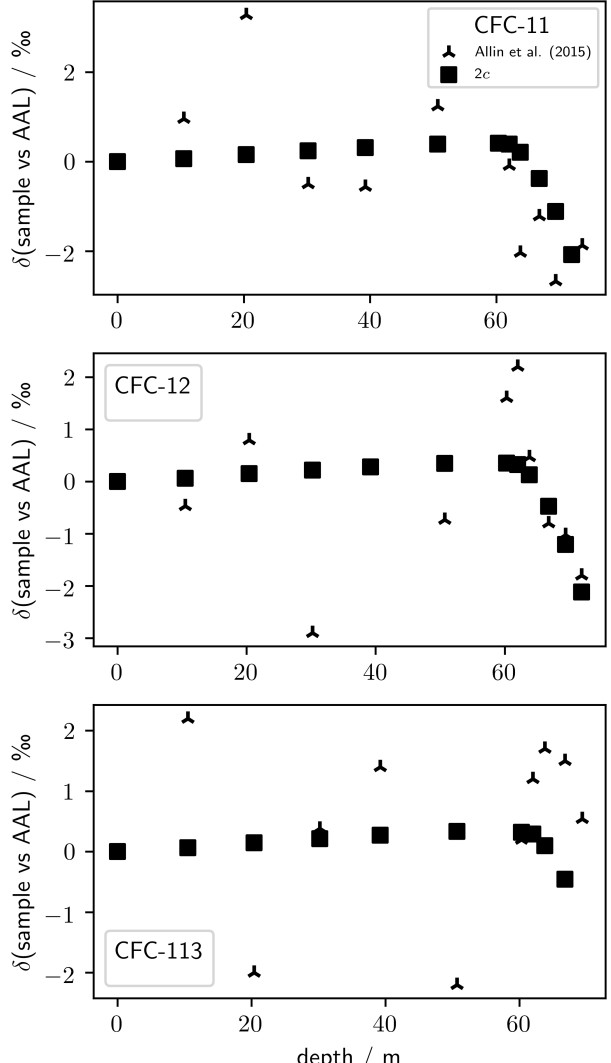

**Figure 1.** $\delta(^{37}\text{Cl})$ as measured by Allin et al. (2015) against depth in NEEM 2009 firn air samples. Also shown is the correction factor, $c$, used in this work to convert $R(102/105)$ to $R(102/101)$ (Equation 2), multiplied by 2. $c$ gives the fractionation for a relative mass difference of 1 and, assuming no change in $\delta(^{37}\text{Cl})$ – which has a relative mass difference of $2 - 2c$ should be consistent with $\delta(^{37}\text{Cl})$.

## 2.5 Modelling the tropospheric isotopic composition

We modelled $\delta_T(^{13}\text{C, CFC-11})$, $\delta_T(^{13}\text{C, CFC-12})$, and $\delta_T(^{13}\text{C, CFC-113})$ from 1937 to 2050 using a two box model. The model was used by Röckmann et al. (2003a), Bernard et al. (2006), and Prokopiou et al. (2017) for $N_2O$ isotopologue budget calculations, and was adopted by Allin et al. (2015) to model the evolution of chlorine isotopes in CFC-11, -12, and -113.
5  This model is detailed in Allin et al. (2015) so we only present a brief overview. The model boxes represent the troposphere and stratosphere. CFCs are emitted to the tropospheric box with a constant isotopic composition, $\delta_E(^{13}\text{C})$. Some portion of

the tropospheric CFC load is transported to the stratospheric box, where CFCs are destroyed and fractionated according to $\epsilon_{app}$. As these fractionated CFCs are exchanged with the troposphere, they alter the tropospheric isotopic composition — our desired variable. Troposphere/stratosphere exchange is parametrised according to Holton (1990) and Appenzeller et al. (1996). The dominant uncertainties in the model are the uncertainty in $\epsilon_{app}$ and the magnitude of the bulk air troposphere/stratosphere

exchange flux, both of which are accounted for in the model uncertainty envelope. We offset modelled $\delta_T(^{13}C)$ such that it is $0\,‰$ in 2005. This treatment ensures that the modelled $\delta_T(^{13}C)$ is relative to the tropospheric composition in 2005, consistent with our data, which are referenced to a 2005 air standard (AAL). We also shift the uncertainty envelope such that it is $0\,‰$ in 2005 and increases backwards and forwards in time, reflecting the fact that in 2005 $\delta_T(^{13}C) = 0\,‰$ by definition. Our only change to the modelling of Allin et al. (2015) is to the value of $\epsilon_{app}$ such that it reflects $^{13}C$ rather than $^{37}Cl$ fractionation.

## 3   Results

### 3.1   Measurements of $\delta(^{13}C)$ in firn air

Measurements of firn air from NEEM 2008/09, plotted against mean age of air, are shown in Figure 2. In the absence of a calibration of our AAL standard against the international standard VPDB, we present $\delta(^{13}C,\,CFC\text{-}11)$ and $\delta(^{13}C,\,CFC\text{-}113)$ relative to our 2005 background air reference gas (AAL). For CFC-12, there were measurements of the same samples on the

VPDB scale (Zuiderweg et al., 2013), allowing us to re-scale our measurements. Taking the mean of the NEEM 2009 samples from 50.7 m to 10.5 m gave $\delta(^{13}C,\,sample\ vs\ VPDB) = (-42.4 \pm 1.4)\,‰$ and $\delta(^{13}C,\,sample\ vs\ AAL) = (0.6 \pm 1.9)\,‰$, resulting in

$$
\delta(^{13}C,\,AAL\ vs\ VPDB) =
$$
$$
\frac{\delta(^{13}C,\,sample\ vs\ AAL) - \delta(^{13}C,\,sample\ vs\ VPDB)}{1 + \delta(^{13}C,\,sample\ vs\ AAL)} = \tag{3}
$$
$$
(43.0 \pm 2.3)\,‰,
$$

which we used to re-scale our $\delta(^{13}C,\,CFC\text{-}12,\,sample\ vs\ AAL)$ measurements to VPDB. A smooth $\delta(^{13}C)$ trend and uncertainty

envelope was calculated using the non-parametric LOESS (locally weighted scatter plot smoothing) technique. Uncertainty in the trend derives from measurement uncertainty and the width of the age distribution at each depth. To account for the age uncertainty, we sub-sampled the relevant probability-weighted age distribution 200 times for each measurement and calculated the LOESS using the resulting measurement pairs. All further details are supplied in the Supplementary Information. The mean standard error on the LOESS was $1.0\,‰$ (CFC-11), $1.3\,‰$ (CFC-12), and $1.6\,‰$ (CFC-113).

We calculated trends in $\delta(^{13}C)$ using the LOESS. For each CFC we saw an increase in $\delta(^{13}C)$ from the oldest to the youngest air: $\delta(^{13}C,\,CFC\text{-}11)$ increased by $(2.9 \pm 1.6)\,‰$ between 1952 and 2009; $\delta(^{13}C,\,CFC\text{-}12)$ increased by $(5.3 \pm 2.2)\,‰$ between 1954 and 2009; and $\delta(^{13}C,\,CFC\text{-}113)$ increased by $(9.3 \pm 2.7)\,‰$ between 1973 and 2009 (standard error and mean age). The observed trends had large relative uncertainty and were similar in magnitude to our *m/z*105 to *m/z*101 correction.

For CFC-12, there was general agreement between our measured firn profile and the measurements of Zuiderweg et al. (2013) after around 1990. However, in the oldest two samples measured by Zuiderweg et al. (2013) – corresponding to mean ages of 1965 and 1977 – there was a significant difference between our measurements of the NEEM 2009 profile and those presented by Zuiderweg et al. (2013). For the 1965 sample, the measurements of Zuiderweg et al. (2013) were around 40

‰ outside of our 95 % confidence intervals, an order of magnitude larger than our $m/z105$ to $m/z101$ correction.

## 3.2 Calculating $\epsilon_{\mathrm{app}}$ from stratospheric measurements

Our stratospheric measurements are presented as Rayleigh plots in Figure 3, where $f$ is the fractional release factor quantifying the degree of stratospheric destruction (Leedham Elvidge et al., 2018). Destruction of CFC-12 and -113 (corresponding to an increase in fractional release factor and decreasing $\ln(1-f)$) was concurrent with an increase in $\delta(^{13}\mathrm{C})$ for the remaining

stratospheric pool (Figure 3). The gradient of the linear regression of $\ln(1+\delta(^{13}\mathrm{C}))$ with $\ln(1-f))$ gives $\epsilon_{\mathrm{app}}$, which was negative for CFC-12 and -113 in both latitude regions. CFC-11 is omitted from Figure 3 because we do not take $\epsilon_{\mathrm{app}}(\mathrm{CFC}\text{-}11)$ forward for our modelling or analysis. We present stratospheric CFC-11 data and justify their omission from our analysis in Appendix D.

From our stratospheric measurements, we derived $\epsilon_{\mathrm{app}}(\mathrm{CFC}\text{-}12, \text{high-lat}) = (-20.2 \pm 4.4)$ ‰ $(p < 0.01)$, $\epsilon_{\mathrm{app}}(\mathrm{CFC}\text{-}12, \text{mid-}$

lat$) = (-30.3 \pm 10.7)$ ‰ $(p = 0.07)$, $\epsilon_{\mathrm{app}}(\mathrm{CFC}\text{-}113, \text{high-lat}) = (-9.4 \pm 4.4)$ ‰ $(p = 0.04)$, and $\epsilon_{\mathrm{app}}(\mathrm{CFC}\text{-}113, \text{mid-lat}) = (-34.4 \pm 9.8)$ ‰ $(p = 0.04)$ — Table 2. Of these, $\epsilon_{\mathrm{app}}(\mathrm{CFC}\text{-}12, \text{mid-lat})$ is significant at 90 % confidence, with the others significant at 95 %.

We derived $\epsilon_{\mathrm{app}}(\mathrm{CFC}\text{-}11)$ by scaling our measured $\epsilon_{\mathrm{app}}(\mathrm{CFC}\text{-}12)$ based on previous laboratory measurements of photolytic carbon isotope fractionation, $\epsilon_{\mathrm{p}}$ (Zuiderweg et al., 2012). $\epsilon_{\mathrm{app}}$ is less than $\epsilon_{\mathrm{p}}$ because atmospheric mixing dilutes the isotopic

effect of photolytic fractionation (Kaiser et al., 2006). Atmospheric mixing affects CFC-11 and -12 similarly so we expect

$$\epsilon_{\mathrm{app}}(\mathrm{CFC}\text{-}11) \approx \epsilon_{\mathrm{app}}(\mathrm{CFC}\text{-}12)\frac{\epsilon_{\mathrm{p}}(\mathrm{CFC}\text{-}11)}{\epsilon_{\mathrm{p}}(\mathrm{CFC}\text{-}12)}. \qquad (4)$$

Taking the mean of $\epsilon_{\mathrm{p}}$ measured at stratospherically relevant temperatures (203 K and 233 K) gives $\epsilon_{\mathrm{p}}(^{13}\mathrm{C}, \mathrm{CFC}\text{-}11) = (-23.4 \pm 0.7)$ ‰ and $\epsilon_{\mathrm{p}}(^{13}\mathrm{C}, \mathrm{CFC}\text{-}12) = (-60.8 \pm 2.2)$ ‰, such that $\frac{\epsilon_{\mathrm{p}}(\mathrm{CFC}\text{-}11)}{\epsilon_{\mathrm{p}}(\mathrm{CFC}\text{-}12)} = 0.39 \pm 0.02$. Scaling our measured $\epsilon_{\mathrm{app}}(\mathrm{CFC}\text{-}$12) by this factor gives $\epsilon_{\mathrm{app}}(^{13}\mathrm{C}, \mathrm{CFC}\text{-}11, \text{high-lat}) = (-7.8 \pm 1.7)$ ‰ and $\epsilon_{\mathrm{app}}(^{13}\mathrm{C}, \mathrm{CFC}\text{-}11, \text{mid-lat}) = (-11.7 \pm 4.2)$ ‰.

These are the best estimates of $\epsilon_{\mathrm{app}}(^{13}\mathrm{C}, \mathrm{CFC}\text{-}11)$ possible using our measurements. For each CFC, $\epsilon_{\mathrm{app}}$ was more negative at mid-latitudes. High-latitude $\epsilon_{\mathrm{app}}$ were derived from more data than the mid-latitude $\epsilon_{\mathrm{app}}$. Of the three CFCs, $\epsilon_{\mathrm{app}}(\mathrm{CFC}\text{-}11)$ was least negative at both latitudes, while $\epsilon_{\mathrm{app}}(\mathrm{CFC}\text{-}12)$ was most negative at high-latitudes and $\epsilon_{\mathrm{app}}(\mathrm{CFC}\text{-}113)$ was most negative at mid-latitudes. We took $\epsilon_{\mathrm{app}}(\text{high-lat})$ forward for our modelling because these were derived from more data and we have more confidence in them.

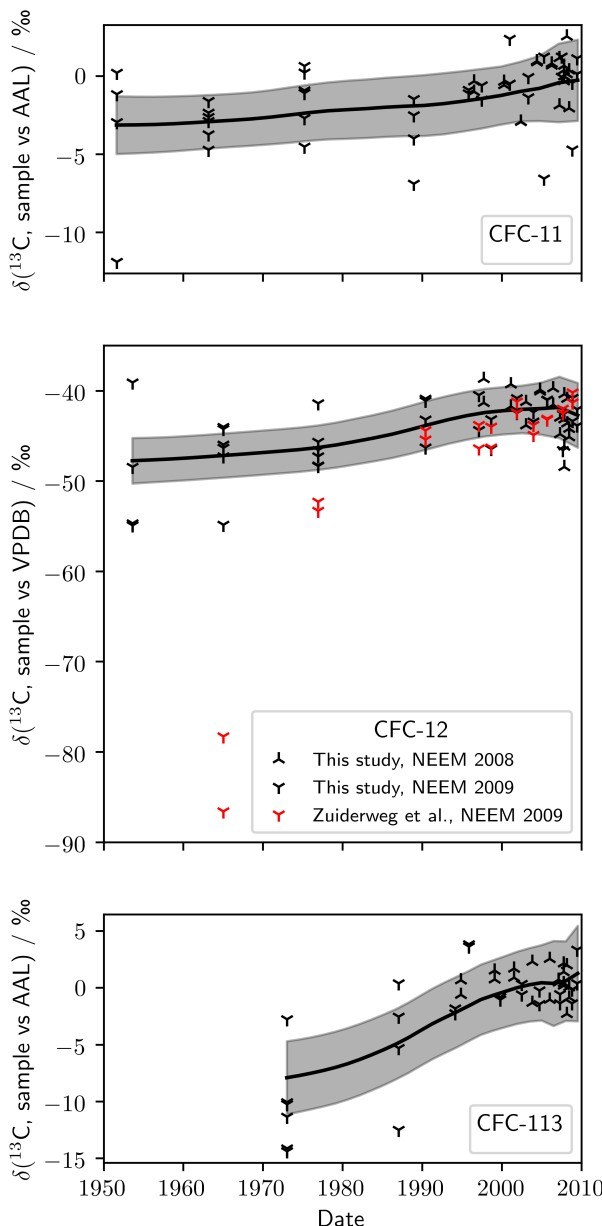

**Figure 2.** Measured $\delta(^{13}$C, CFC-11) [top], $\delta(^{13}$C, CFC-12) [middle], and $\delta(^{13}$C, CFC-113) [bottom] in NEEM 2008 and 2009 firn air. Also shown is $\delta(^{13}$C, CFC-12) as measured by Zuiderweg et al. (2013) in the same NEEM 2009 firn air samples. The smoothed trend (black line) and 95% confidence bounds (grey shading) were generated using a LOESS regression.

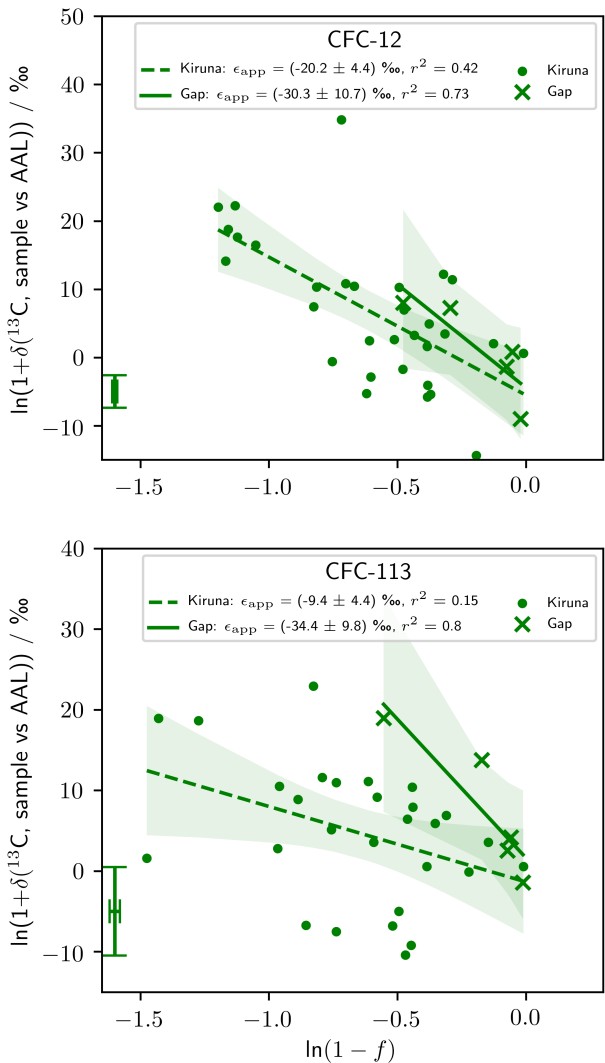

**Figure 3.** Rayleigh plots of our stratospheric measurements. The linear regression (lines) and 95 % confidence bounds on the regression (shading) are shown for the high-latitude (Kiruna) and mid-latitude (Gap) data sets. The gradients of these regressions, corresponding to $\epsilon_{app}$, are given in the legend with one standard error. The errorbar in the bottom left corner of each graph shows the median repeatability of the reference gas measurements over the measurement days and the median error deriving from the fractional release factor. CFC-11 data are presented in Appendix D.

### 3.3 Reconstructed tropospheric isotopic composition

We now turn to our measured and modelled $\delta_T(^{13}C)$ (Figure 4). Our firn measurements have been corrected for gravitational and diffusive fractionation. The smoothed trend and 95 % confidence interval were, similar to the firn profiles, based on LOESS regression on these corrected data. The standard error on the $\delta_T(^{13}C)$ reconstruction was equal to that of the firn profile to within 0.1 ‰ for each CFC. Modelled $\delta_T(^{13}C)$ is presented with 95 % confidence intervals. The model was forced with

**Table 2.** The apparent isotopic fractionation, $\epsilon_{app}$, derived from our stratospheric measurements and the photolytic isotopic fractionation, $\epsilon_p$, measured by Zuiderweg et al. (2012). All uncertainties are one standard error.

| CFC | $\epsilon_{app}$ / ‰ | | $\epsilon_p$ / ‰ | |
| --- | --- | --- | --- | --- |
| | High-latitude | Mid-latitude | 203 K | 233 K |
| 11 | $-7.8 \pm 1.7^\dagger$ | $-11.7 \pm 4.2^\dagger$ | $-23.8 \pm 0.9$ | $-23.0 \pm 1.1$ |
| 12 | $-20.2 \pm 4.4$ | $-30.3 \pm 10.7$ | $-66.2 \pm 3.1$ | $-55.3 \pm 3.0$ |
| 113 | $-9.4 \pm 4.4$ | $-34.4 \pm 9.8$ | | |

$^\dagger$Calculated using Equation 4. Taking the mean of the 203 K and 233 K measurements gives $\frac{\epsilon_{p(CFC\text{-}11)}}{\epsilon_{p(CFC\text{-}12)}} = 0.39 \pm 0.02$.

our derived $\epsilon_{app}$(high-lat), prescribed CFC emissions, and a constant isotopic composition of emissions. For CFC-12, we also show the polynomial presented by Zuiderweg et al. (2013) representing the tropospheric trend that best captured their firn measurements.

As with the firn profile, we calculated trends in measured $\delta_T(^{13}C)$ using a LOESS. For each CFC, measured $\delta_T(^{13}C)$ in-creased through time: $\delta_T(^{13}C, CFC\text{-}11)$ increased by $(2.1 \pm 1.6)$ ‰ between 1952 and 2009; $\delta_T(^{13}C, CFC\text{-}12)$ increased by $(4.8 \pm 2.2)$ ‰ between 1956 and 2009; and $\delta_T(^{13}C, CFC\text{-}113)$ increased by $(9.0 \pm 2.7)$ ‰ between 1975 and 2009 (standard error and mean age). These trends are similar to, and slightly smaller than, the trends in the firn because of the gravitational/diffusive correction. The polynomial of Zuiderweg et al. (2013) only agreed with our reconstructed $\delta_T(^{13}C, CFC\text{-}12)$ after around 1995. Our measurements are compared to previously published $\delta_T(^{13}C, CFC\text{-}11)$, $\delta_T(^{13}C, CFC\text{-}12)$, and $\delta_T(^{13}C, CFC\text{-}113)$ in Table 3. Our $\delta_T(^{13}C, CFC\text{-}12)$ measurements were re-scaled to Zuiderweg et al. (2013) using 50.7 m and shallower samples rep-resentative of mean ages of around 2000 to 2009. In this period, our measured $\delta_T(^{13}C, CFC\text{-}12)$ is therefore consistent with Zuiderweg et al. (2013) by definition. These measurements were consistent with Bahlmann et al. (2011) $(-41.2 \pm 0.2)$ ‰ and with Redeker et al. (2007) $(-40.3 \pm 2.6)$ ‰ to within one standard deviation. For each CFC, our $\delta_T(^{13}C,$ sample vs AAL) for this period was consistent with 0 ‰, as expected given our AAL reference was collected in 2005. For CFC-11 and -113, a quantitative comparison of our data to previous measurements (Thompson et al., 2002; Redeker et al., 2007; Bahlmann et al., 2011) was not possible due to our data being on a different scale.

We take the mean of the $\delta_T(^{13}C)$ as predicted using the Appenzeller et al. (1996) and Holton (1990) stratosphere/troposphere exchange parametrisations for a given emissions scenario and $\epsilon_{app}$ to be one model scenario. Hence, four model scenarios are shown in Figure 4: one for each CFC using Velders and Daniel (2014) emissions (three scenarios, labelled V&D); plus an additional scenario for CFC-11 that differs from V&D by fixing emissions after 2012 (one scenario, labelled M18). The model uncertainty for a scenario is taken to be the full $\delta_T(^{13}C)$ envelope as predicted using the two stratosphere/troposphere exchange parametrisations. The dominant uncertainty was from $\epsilon_{app}$, as shown by the strong overlap between the uncertainty envelopes of the two stratosphere/troposphere exchange parametrisations. Each model run predicted an increase in $\delta_T(^{13}C)$ through time. This behaviour is qualitatively consistent with our measurements for each CFC. There was quantitative agreement between our measurements and modelling for CFC-11 and CFC-12 for the entire period covered by the measurements. For CFC-113,

**Table 3.** Comparison of $\delta_T(^{13}C)$ measurements for CFC-11, -12, and -113 from various studies. Only measurements from background, rural, or coastal sites are included as these are most representative of the remote firn measurements presented here. The uncertainties are one standard deviation and the number in brackets gives the number of observations.

| | $\delta_T(^{13}C$, sample vs VPDB$)$ / ‰ | | | | | $\delta_T(^{13}C$, sample vs AAL$)$ / ‰ |
| --- | --- | --- | --- | --- | --- | --- |
| | Thompson et al. (2002)[†] | Redeker et al. (2007)[*] | Bahlmann et al. (2011)[+] | Zuiderweg et al. (2013)[‡] | This study[§] | This study[§] |
| CFC-11 | | $-26.8 \pm 4.4$ (9) | $-31.5 \pm 2.6$ (3) | | | $-0.5 \pm 1.9$ (31) |
| CFC-12 | | $-40.3 \pm 2.6$ (9) | $-41.2 \pm 0.2$ (3) | $-42.5 \pm 1.4$ (10) | $-42.5 \pm 2.2$ (31) | $0.5 \pm 2.3$ (31) |
| CFC-113 | $-23.3 \pm 9.6$ (38) | $-12.6 \pm 6.8$ (1) | $-25.4 \pm 1.1$ (3) | | | $0.2 \pm 1.4$ (31) |

[†]Thompson et al. (2002), sampled July 1999 to March 2001 background northern (Alert, Canada (82.5 °N, 62.3 °W)) and southern hemisphere (Baring Head, New Zealand (41.4 °S, 174.9 °E)) air

[*]Redeker et al. (2007), sampled August to November 2004, Crossgar (54.40 °N, 5.76 °W), Hillsborough (54.46 °N, -6.08 °W), and Mace Head (53.20 °N, 9.54 °W)

[+]Bahlmann et al. (2011), marine influenced air, Wadden Sea Station in List/Sylt (55.02 °N, 8.44 °E), August to September 2010

[‡]Zuiderweg et al. (2013), firn air samples representing, depths 50.7 m and shallower, representing 2002 to 2009; NEEM 2009 campaign

[§]This study, firn air samples corrected for gravitational and diffusive fractionation, depths 50.7 m and shallower, representing: 2001 to 2009 (CFC-11), 2002 to 2009 (CFC-12), 2000 to 2009 (CFC-113); NEEM 2008/09 campaigns

the model is consistent with the measurements after around 1980, but predicts too little fractionation to capture the observed $\delta_T(^{13}C)$ depletion measured in the sample with a mean age of 1975.

Our model returned a value for the isotopic composition of emissions, $\delta_E(^{13}C)$, such that modelled $\delta_T(^{13}C) = 0$ ‰ in 2005 (Table 4). Hence, more negative $\epsilon_{app}$, which drive greater fractionation up to 2005, produce more negative $\delta_E(^{13}C)$. We
take $\delta_E(^{13}C)$ as the mean of the predictions using the Appenzeller et al. (1996) and Holton (1990) stratosphere/troposphere exchange parameterisations. For each CFC, $\delta_E(^{13}C)$ was negative and significantly different from 0 ‰, relatively depleted in $^{13}C$ compared to 2005 tropospheric air. For CFC-12, we can perform a quantitative comparison with previously reported $\delta(^{13}C$, sample vs VPDB$)$ values of CFC-12 gas that was purchased from manufacturers, which range from $-33$ ‰ (Ertl, 1997) to $(-46.8 \pm 0.2)$ ‰ (Archbold et al., 2012) (Table 4). We modelled $\delta_E(^{13}C$, CFC-12$) = (-47.1 \pm 1.3)$ ‰ (two standard errors),
within the range of previously reported $\delta_E(^{13}C$, CFC-12$)$.

## 4   Discussion

Our measurements provide the first observational constraints on $\epsilon_{app}(^{13}C)$ for CFC-12 and -113. Our derived $\epsilon_{app}(^{13}C$, CFC-12$)$ are consistent with previously reported $\epsilon_p(^{13}C$, CFC-12$)$ (Zuiderweg et al., 2012), being a factor of 2 to 3 lower than $\epsilon_p(^{13}C$, CFC-12$)$, as expected given the effect of mixing and diffusion in the atmosphere (Kaiser et al., 2006). Kaiser et al.
(2006) also discuss in detail the effects of mixing and transport on the apparent isotope fractionations for $N_2O$ (which we know with much better precision). While mixing and transport are certainly relevant and reduce the observed apparent stratospheric isotope fraction compared with the intrinsic photochemical isotope effects, the variations in these mixing and transport effects are negligible for the precision that we report for CFCs, as can be inferred from the more precise observations for $N_2O$. The meridional differences observed in $\epsilon_{app}($CFC-12$)$ are qualitatively consistent with previously reported isotopic fractionation
patterns of other elements in long-lived trace gases, such as $\delta(^{37}Cl$, CFC-12$)$ (Allin et al., 2015; Laube et al., 2010b) and $N_2O$ isotopologues (Kaiser et al., 2006). For CFC-113, we have only measured the $\delta(^{13}C)$ of the $CCl_2F$ fragment and our results do not provide information on the $CClF_2$ fragment or for the molecule as a whole. Our derived $\epsilon_{app}($CFC-113$)$ are internally consistent with our firn air measurements, with both data sets being measured on the same fragment. Allin et al.

**Table 4.** $\delta_E(^{13}C)$ as predicted by our modelling and as reported in previous studies for CFC-11, -12, and -113. With the exception of Ertl (1997), all uncertainties are two standard errors.

| | $\delta_E(^{13}C,$ sample vs AAL$)/\permil$ | $\delta_E(^{13}C,$ sample vs VPDB$)/\permil$ | | | | | |
|---|---|---|---|---|---|---|---|
| | This study | This study | Ertl (1997) | Thompson et al. (2002) | Archbold et al. (2005) | Horst et al. (2015) | Phillips et al. (2019) |
| CFC-11 | $-2.7 \pm 0.8$ | | $-35$ to $-25$ | | $-26.2 \pm 0.6$ | $-33.34 \pm 0.07$<br>$-28.93 \pm 0.04$ | $-33.36 \pm 0.04$ |
| CFC-12 | $-4.3 \pm 1.3$ | $-47.1 \pm 1.3$ | $-45$ to $-33$ | | $-46.8 \pm 0.2$ | | $-45.27 \pm 0.04$ |
| CFC-113 | $-1.7 \pm 1.1$ | | | $-31.3 \pm 0.5$ | $-26.5 \pm 0.8$ | $-28.07 \pm 0.05$ | $-29.93 \pm 0.06$ |

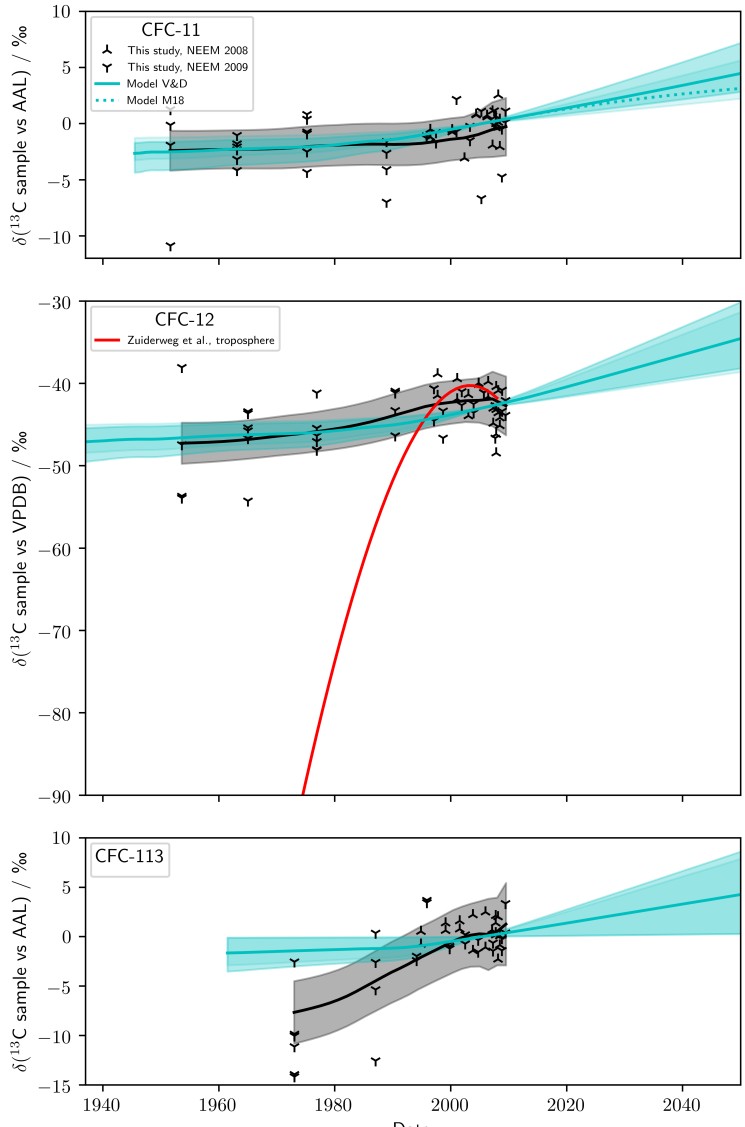

**Figure 4.** Measured and modelled $\delta_T(^{13}C, CFC-11)$, $\delta_T(^{13}C, CFC-12)$, and $\delta_T(^{13}C, CFC-113)$. The smoothed trend and 95 % confidence intervals for measured $\delta_T(^{13}C)$ are shown by the solid black line and grey shading, respectively. The solid blue line shows the mean modelled $\delta_T(^{13}C)$ for the Velders and Daniel (2014) scenario (V&D) and the two stratosphere/troposphere exchange parameterisations (Holton, 1990; Appenzeller et al., 1996). $\epsilon_{app}$(high-lat) was used for each CFC. Blue shading shows the 95 % uncertainty envelope for the Holton (1990) and Appenzeller et al. (1996) parameterisations. For CFC-11 only, the dotted blue line shows the mean modelled $\delta_T(^{13}C)$ for the M18 scenario, with the uncertainty envelope omitted for clarity. For CFC-12, the red line shows $\delta_T(^{13}C)$ as predicted by Zuiderweg et al. (2013) [amending a typo in the coefficients presented by Zuiderweg et al. (2013), $\delta_T(^{13}C, CFC-12) = -265.4280 + 4.8315x + (7.8555 \times 10^{-5})x^2 - (3.3070 \times 10^{-4})x^3$, where $x = t - 1933.5$ and $t$ gives the date].

(2015) do not observe meridional differences for $\epsilon_{app}(^{37}Cl, CFC-113)$ – as we observe for $\epsilon_{app}(^{13}C, CFC-113)$ – but speculate that differences could be masked by their uncertainties. Given our best understanding of compact tracer-tracer correlations in

the lower stratosphere we do not expect significant meridional differences in $\epsilon_{\mathrm{app}}$ for the range of observed fractional releases ($\ln(1-f) > -0.6$)) (Volk et al., 1997). The lack of a latitude-dependence for lower-stratospheric Rayleigh fractionation is supported by stratospheric observations of other long-lived trace gases, in particular carbon and hydrogen isotope fractionation in $CH_4$ (Röckmann et al., 2011) and nitrogen and oxygen isotope fractionation in $N_2O$ (Kaiser et al., 2006), which are con-
strained by a much wider range of observations, with lower measurement uncertainties, than currently available for CFCs. $CH_4$ and $N_2O$ have global atmospheric mean lifetimes of 10 years and 123 years, respectively, which covers the range of lifetimes of CFC-11 (52 years), CFC-113 (93 years) and CFC-12 (102 years). The three CFCs also have the same chemical sinks as $N_2O$ – photolysis and oxidation by $O(^1D)$, in similar proportions as $N_2O$. We therefore do not expect these CFCs to behave any differently than $CH_4$ and $N_2O$. The observed meridional differences could be statistical artefacts deriving from our poorly
constrained $\epsilon_{\mathrm{app}}$(mid-lat).

    We derived $\epsilon_{\mathrm{app}}$(CFC-11) by scaling our measured $\epsilon_{\mathrm{app}}$(CFC-12) to previously reported $\frac{\epsilon_{\mathrm{p}}(\text{CFC-11})}{\epsilon_{\mathrm{p}}(\text{CFC-12})}$ (Zuiderweg et al., 2012). While our presented $\epsilon_{\mathrm{app}}$(CFC-11) are our best estimates, they are dependent on our $\epsilon_{\mathrm{app}}$(CFC-12) estimates, $\epsilon_{\mathrm{p}}$(CFC-11), and $\epsilon_{\mathrm{p}}$(CFC-12). Our re-scaled $\epsilon_{\mathrm{app}}$(CFC-11) also omit the effect of fractionation by $O(^1D)$, though given the around 2 % contribution of $O(^1D)$ to stratospheric CFC-11 loss (Burkholder et al., 2013), we expect this omission to have little effect.
Appendix D gives technical details on why we do not use our CFC-11 measurements in our analysis and modelling. For all three CFCs, our $\epsilon_{\mathrm{app}}$(high-lat) were derived from more data than our $\epsilon_{\mathrm{app}}$(mid-lat) so we have more confidence in our $\epsilon_{\mathrm{app}}$(high-lat). For each CFC, $\epsilon_{\mathrm{app}}(^{13}C)$ is larger than previously reported $\epsilon_{\mathrm{app}}(^{37}Cl)$ (Allin et al., 2015), causing more negative fractionation during stratospheric destruction.

    Our tropospheric reconstructions and modelling (Figure 4) allow us to investigate changes in $\delta_{\mathrm{E}}(^{13}C)$. Our model was
run using a constant $\delta_{\mathrm{E}}(^{13}C)$ and agreement between our reconstructed and modelled $\delta_{\mathrm{T}}(^{13}C)$ is therefore evidence that no large change in $\delta_{\mathrm{E}}(^{13}C)$ has occurred over the time period spanned by the measurements. For CFC-12, there was agreement between our reconstructed $\delta_{\mathrm{T}}(^{13}C)$ and comparable previous measurements (Table 3). This agreement reflects the calibration of Zuiderweg et al. (2013), to which our measurements were re-scaled, but is still a check on the quality of our reconstruction. Our modelled and measured $\delta_{\mathrm{T}}(^{13}C, \text{CFC-12})$ were in agreement for the entire period covered by the measurements (Figure
4). Our results are therefore consistent with a constant $\delta_{\mathrm{E}}(^{13}C, \text{CFC-12})$. Furthermore, $\delta_{\mathrm{E}}(^{13}C, \text{CFC-12})$ as predicted by our model was within the range of previously reported isotopic source compositions for CFC-12 (Table 4). While some variation in $\delta_{\mathrm{E}}(^{13}C, \text{CFC-12})$ is possible within our uncertainties, these confluent lines of evidence suggest that no dramatic change in $\delta_{\mathrm{E}}(^{13}C, \text{CFC-12})$, as proposed by Zuiderweg et al. (2013), has occurred since around 1956. The cause of this discrepancy was likely an analytical artefact in Zuiderweg et al. (2013), discussed further in Appendix C. Our measurements and modelling of
$\delta_{\mathrm{T}}(^{13}C, \text{CFC-11})$ are in agreement for the entire period covered by measurements and are therefore consistent with a constant $\delta_{\mathrm{E}}(^{13}C, \text{CFC-11})$ since at least 1952. For CFC-113, our modelling did not agree with our measurements earlier than around 1980. This discrepancy may be indicative of a change in $\delta_{\mathrm{E}}(^{13}C, \text{CFC-113})$ though, given our measurements do not provide a complete picture of the fractionation in CFC-113 and given this discrepancy is caused by one measurement depth, these results do not confirm a change in $\delta_{\mathrm{E}}(^{13}C, \text{CFC-113})$. Multiple industrial processes use CFC-113 as a feedstock, or produce
CFC-113 as an intermediate (Adcock et al., 2018), so a change in $\delta_{\mathrm{E}}(^{13}C, \text{CFC-113})$ is plausible. The discrepancy between the

modelled and reconstructed confidence bounds is at most -1.9 ‰. The range of published $\delta_E$(CFC-113) is $(-31.3 \pm 0.5)$ ‰ to $(-26.5 \pm 0.8)$ ‰ (Table 4), around 5 ‰. The discrepancy seen for measured and modelled $\delta_T$(CFC-113) can be accounted for by the range of published $\delta_E$(CFC-113).

Our modelling predicts increasing $\delta_T(^{13}C)$ for CFC-11, -12, and -113, as lighter isotopologues are preferentially destroyed in the stratosphere and the remaining stratospheric CFC pool, enriched in $^{13}C$, is mixed with tropospheric air. An acceleration in the rate of increase of $\delta_T(^{13}C)$ was modelled for each CFC, starting in around 1990. This acceleration is caused by reduced emissions, with relatively depleted $\delta_E(^{13}C)$, as emissions mitigate stratospheric $^{13}C$ enrichment. Therefore, the new CFC-11 emissions identified by Montzka et al. (2018) have the potential to decrease the rate of increase in $\delta_T(^{13}C$, CFC-11). We estimated the potential effect of these new emissions by comparing the V&D and M18 scenarios. As expected, in M18, $\delta_T(^{13}C$, CFC-11) was lower than V&D after 2012. Using $\epsilon_{app}$(CFC-11, high-lat), the difference was 1.4 ‰ in 2050 — well within our uncertainty envelope. Improved modelling precision and more precise knowledge of $\delta_E(^{13}C$, CFC-11) would be needed if $\delta_T(^{13}C$, CFC-11) measurements were to be used as a tool for monitoring global CFC-11 emissions, though the isotopic signal from emissions may be more pronounced on regional scales.

## 5 Conclusions

We presented a new dataset of the $\delta(^{13}C)$ of CFC-11, -12, and -113 for stratospheric air samples, and derived values for the apparent isotopic fractionation, $\epsilon_{app}$, at high- and mid-latitudes of: $\epsilon_{app}$(CFC-11, high-lat) $= (-7.8 \pm 1.7)$ ‰; $\epsilon_{app}$(CFC-11, mid-lat) $= (-11.7 \pm 4.2)$ ‰; $\epsilon_{app}$(CFC-12, high-lat) $= (-20.2 \pm 4.4)$ ‰; $\epsilon_{app}$(CFC-12, mid-lat) $= (-30.3 \pm 10.7)$ ‰; $\epsilon_{app}$(CFC-113, high-lat) $= (-9.4 \pm 4.4)$ ‰; and $\epsilon_{app}$(CFC-113, mid-lat) $= (-34.4 \pm 9.8)$ ‰. While for CFC-12 and -113 these estimates are independent, the $\epsilon_{app}$(CFC-11) estimates are not, having been derived by scaling our $\epsilon_{app}$(CFC-12) measurements. Further measurements of $\delta(^{13}C$, CFC-11) in the stratosphere are required to estimate $\epsilon_{app}$(CFC-11) independent of CFC-12. For CFC-113, these $\epsilon_{app}$ are only applicable to the $CCl_2F$ fragment of the molecule. When used to model the tropospheric isotopic composition, $\delta_T(^{13}C)$, our derived $\epsilon_{app}$(high-lat) drive strong fractionation from the mid 1900s through to 2050. Our stratospheric data complement previous, similar measurements of $\delta(^{37}Cl)$ for CFC-11, -12, and -113 (Allin et al., 2015; Laube et al., 2010a).

We also reconstructed $\delta_T(^{13}C)$ from firn air measurements. Comparing these with the model shows that the histories of $\delta_T(^{13}C$, CFC-11) and $\delta_T(^{13}C$, CFC-12) are consistent with a constant isotopic source composition, $\delta_E(^{13}C)$, and with stratospheric processing as the sole sink of these chemicals. Our results contradict previous reports of extreme depletion for $\delta_T(^{13}C$, CFC-12) and $\delta_E(^{13}CFC$-12). Such extreme depletions could have challenged the history of CFC-12 industrial processes and feedstocks; the current understanding of their atmospheric cycling; and/or raised questions about their inertness in the biogeosphere. The discrepancy between reconstructed and modelled $\delta_T(^{13}C$, CFC-113) suggests a change in $\delta_E(^{13}C$, CFC-113). Changes in industrial processes that produce CFC-113 – as an end product or byproduct – could explain such a discrepancy, and the range of reported $\delta_E(^{13}C$, CFC-113) would be sufficient to cause such a discrepancy. We caution, however, that this discrepancy derives from only one sample and takes into account the fractionation of only one CFC-113 fragment. Further

work would be needed to definitively assign a change in $\delta_{\mathrm{E}}(^{13}\mathrm{C}, \mathrm{CFC}\text{-}113)$. The modelled increase in $\delta_{\mathrm{T}}(^{13}\mathrm{C})$ from 2009 through 2050 is sensitive to new emissions. We compared future $\delta_{\mathrm{T}}(^{13}\mathrm{C}, \mathrm{CFC}\text{-}11)$ trends in scenarios with/without new CFC-11 emissions. The difference between scenarios was within uncertainty bounds, showing better modelling precision and precise quantification of the isotopic composition of emissions would be needed to detect the isotopic signature of recently reported

new CFC-11 emissions in background air.

*Code and data availability.*  All data and plot scripts used in this study are given as supplementary information.

## Appendix A:  Dilution series and quality control

We measured two dilution series to account for any errors or biases that may be introduced by the low concentrations of CFCs in some samples (Figure A1). The first dilution series was produced using method A, and is therefore applicable to measurements

of the Kiruna and Gap samples. The second dilution series was produced using method B, and is therefore applicable to measurements of the NEEM 2008/09 firn samples. Each dilution series included repeat measurements of a reference gas (unpolluted tropospheric air collected in 2009; SX-0706077) at five concentrations ranging from: $(2.9 \pm 0.02)$ pmol/mol to $(245.1 \pm 3.6)$ pmol/mol (CFC-11); $(6.6 \pm 0.1)$ pmol/mol to $(540.0 \pm 3.4)$ pmol/mol (CFC-12); and $(0.9 \pm 0.01)$ pmol/mol to $(78.1 \pm 0.2)$ pmol/mol (CFC-113). Measurements of this dilution series were previously reported by Allin et al. (2015, SI).

The $m/z\,102$ peak area was used as an indicator of the level of dilution in the sample. Noting that the true $\delta(^{13}\mathrm{C}, \text{sample vs}$ SX-0706077) value of each measurement is 0 ‰, we assessed the performance of our method by plotting peak area against measured $\delta(^{13}\mathrm{C}, \text{sample vs SX-0706077})$ for each sample.

For method A, the measured $\delta(^{13}\mathrm{C})$ was negative for the samples with the lowest $m/z\,102$ peak area. We have therefore taken the lowest dilution series $m/z\,102$ peak area where we do not see this behaviour to be the lower $m/z\,102$ peak area limit

for method A, above which we have reliable data. This limit was 39000 (CFC-11), 57000 (CFC-12), and 44000 (CFC-113), and is shown in Figure A1 by the red dotted line. Kiruna and Gap measurements with $m/z\,102$ peak areas below this threshold were excluded from our results but are provided in the Supplementary Information. For method B, the measured $\delta(^{13}\mathrm{C})$ showed no bias for lower peak areas and we therefore retained all data. We have excluded the 69.4 m and 71.9 m NEEM 2009 samples for CFC-113 because, for the corresponding mean ages, there was too little CFC-113 in the atmosphere (Adcock et al., 2018)

to reliably determine $\delta(^{13}\mathrm{C})$.

## Appendix B:  Comparison of GC-MS with GC-IRMS measurements

Single detector GC-MS has been used previously to measure carbon isotope ratios (Eiler et al., 2017; Hauri et al., 2002; Schutten et al., 1957; Nier, 1940), and our single detector GC-MS analytical system has been used previously to measure chlorine isotopes in stratospheric and firn air samples (Allin et al., 2015; Laube et al., 2010a). As a check on the quality of our

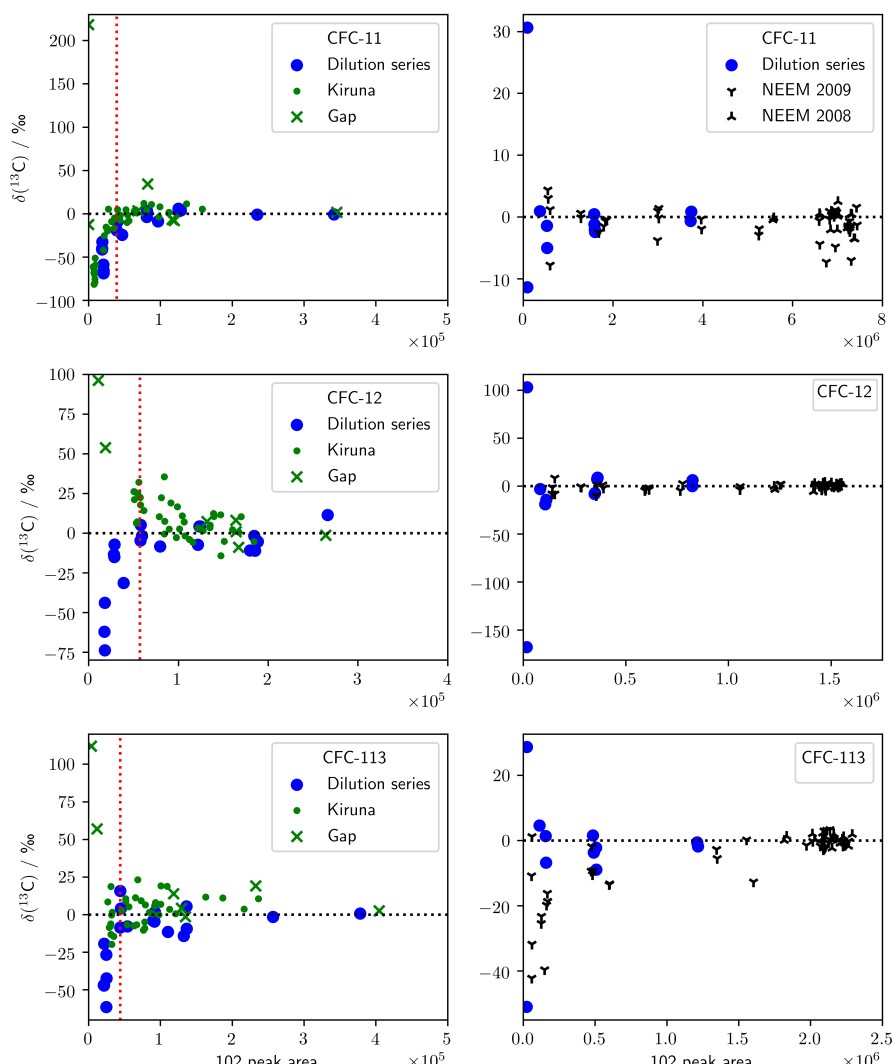

**Figure A1.** Dilution series for the analytical methodologies used in this work. Left: Method A, which was used to measure stratospheric samples. The red dotted line shows the lower limit of $m/z\,102$ peak areas that were retained. Right: Method B, which was used to measure firn air. Dilution series measurements are relative to SX-0706077 (2009 air) and stratospheric and firn air measurements are relative to AAL (2005 air).

method, we have compared measurements made using GC-IRMS (Zuiderweg et al., 2011) of a suite of photolysis samples as presented in Zuiderweg et al. (2012) to our own measurements – using GC-MS – of those samples (Figure B1). These samples were not subject to the $CH_3Cl$ chromatographic interference seen in Zuiderweg et al. (2013). The samples were diluted by 1000 times before measurement on our system to accommodate the higher sensitivity of our GC-MS method.

5    The agreement between the methods is good (Figure B1). Linear regression gives a high regression coefficient ($r^2 = 0.92$) and a gradient consistent with unity ($1.0 \pm 0.1$). This agreement holds over a range of $\delta$ spanning almost 60 ‰. The intercept

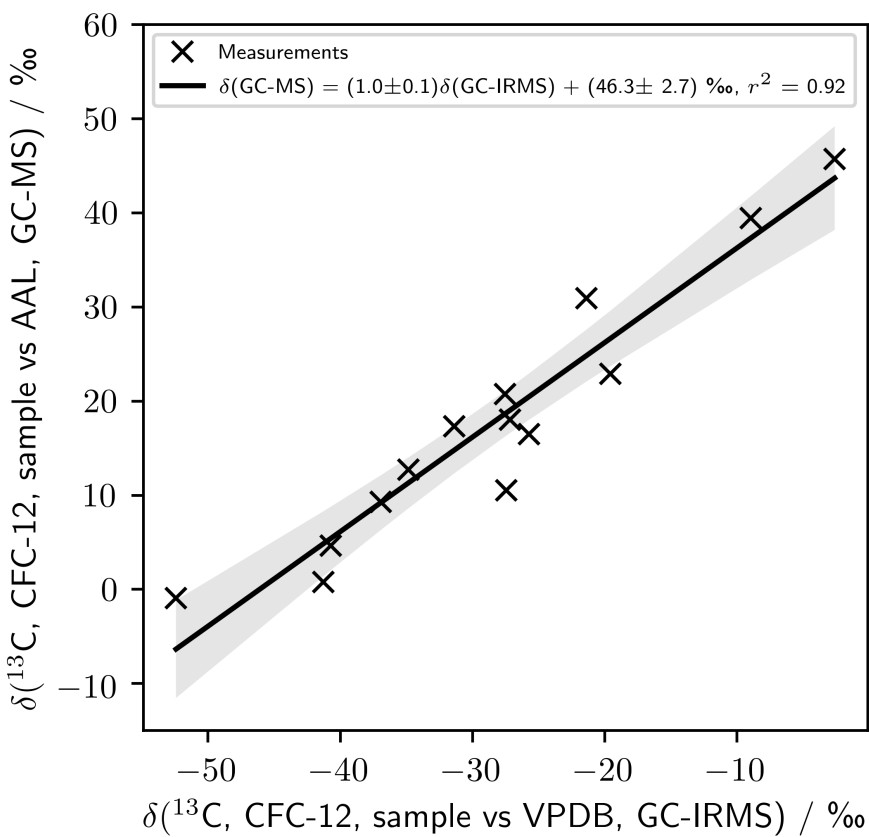

**Figure B1.** Comparison of measurements presented by Zuiderweg et al. (2012) using GC-IRMS (Zuiderweg et al., 2011) to our own measurements of those same samples.

is $(46.3 \pm 2.7)\,‰$. This intercept results in a $\delta$ value for CFC-12 in AAL-071170 on the VPDB scale of $(-44.2 \pm 2.5)\,‰$ — consistent with the $(43.0 \pm 2.3)\,‰$ derived using Equation 3.

### Appendix C: Reason for NEEM 2009 discrepancy

We measured $\delta(^{13}\mathrm{C}, \mathrm{CFC}\text{-}12)$ in the same NEEM 2009 flask samples as Zuiderweg et al. (2013) and linked our measurements to the VPDB calibration scale used by Zuiderweg et al. (2013). The measurements in these two studies were consistent, except for the samples at 66.8 and 69.4 m, corresponding to mean ages of 1977 and 1965, respectively (Figure 2). For the sample corresponding to a mean age of 1965, the discrepancy between data sets is around $40\,‰$, an order of magnitude larger than our 95 % confidence intervals and any corrections made to our measurements. The discrepancy is larger when $\delta_\mathrm{T}(^{13}\mathrm{C}, \mathrm{CFC}\text{-}12)$ is considered, with the tropospheric scenario presented by Zuiderweg et al. (2013) predicting $\delta_\mathrm{T}(^{13}\mathrm{C}) = -123\,‰$ in 1965,

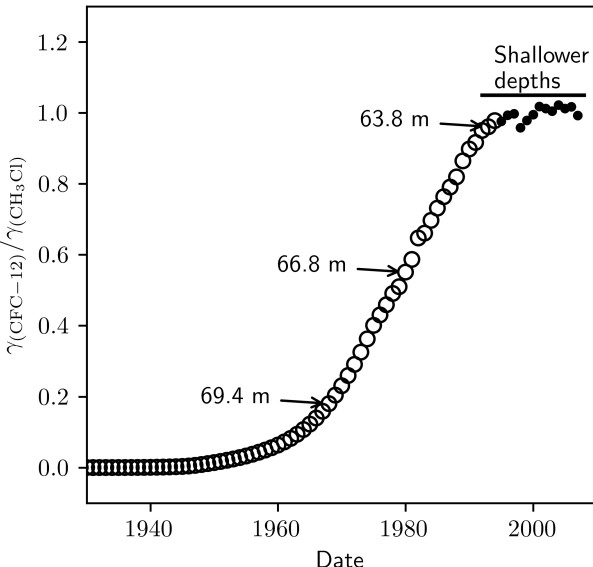

**Figure C1.** The ratio of the tropospheric mole fraction of CFC-12 to $CH_3Cl$. Circles were taken from Butler et al. (1999). The dots were calculated using data from NOAA-HATS. The annotations indicate the points on the curve that correspond to different depth samples in the NEEM 2009 firn profile.

whereas our measured $\delta_T(^{13}C, CFC-12) = (-46.8 \pm 2.4)$ ‰ (2 standard errors). Assuming sample integrity was preserved between studies, at most one data set can be accurate.

The cause of this discrepancy was likely a measurement artefact in Zuiderweg et al. (2013). In the method of Zuiderweg et al. (2013), methyl-chloride elutes before CFC-12. Zuiderweg et al. (2013) model the methyl-chloride tail using an exponentially
decaying function, and subtract this signal from their CFC-12 peak before integration. Zuiderweg et al. (2013) performed a dilution series to evaluate their method, including their treatment of the methyl-chloride peak. However, in their dilution series the proportion of methyl-chloride to CFC-12 was constant because methyl-chloride and CFC-12 were diluted concurrently. In the NEEM 2009 firn air samples, the proportion of methyl-chloride to CFC-12 increased in the deeper samples (see Figure 2 in Zuiderweg et al. (2013)). This increase reflects the changing ratio of the CFC-12 mole fraction, $\gamma(CFC-12)$, to the $CH_3Cl$
mole fraction, $\gamma(CH_3Cl)$, since the early 1900s (Figure C1). The methyl-chloride baseline correction was therefore performed, on the sample at 69.4 m – and, to a lesser extent, on the sample at 66.8 m – on a methyl chloride peak that was larger than that evaluated in the dilution series for a given CFC-12 peak area. Indeed, the trend in $\delta(^{13}C)$ depletion in the NEEM 2009 profile was qualitatively similar to the trend in tropospheric $\gamma(CFC-12)/\gamma(CH_3Cl)$ over that time period, with lower $\gamma(CFC-12)/\gamma(CH_3Cl)$ for the depleted $\delta(^{13}C)$ measurements, and relatively constant $\gamma(CFC-12)/\gamma(CH_3Cl)$ in the period of
little change in $\delta(^{13}C)$. The dilution series performed by Zuiderweg et al. (2013) therefore did not adequately assess variations in $\gamma(CFC-12)/\gamma(CH_3Cl)$.

## Appendix D: Stratospheric CFC-11

We presented $\epsilon_{app}$(CFC-11) based on scaling of our measured $\epsilon_{app}$(CFC-12) (Table 2, Equation 4). These best estimate values were used in our analysis and modelling. We took this approach because the $\epsilon_{app}$(CFC-11) derived from our stratospheric measurements were inadequate. Figure 3 shows stratospheric Rayleigh plots for our CFC-11 data, from which we derived

$\epsilon_{app}$(CFC-11, high-lat) $= (-3.8 \pm 4.9)$ ‰ and $\epsilon_{app}$(CFC-11, mid-lat) $= (-26.5 \pm 4.0)$ ‰ (one standard error). $\epsilon_{app}$(CFC-11, high-lat) is not significantly different from 0 ‰ and therefore, when used to force our model, gives confidence intervals that, while consistent with our observations, span 0 ‰. $\epsilon_{app}$(CFC-11, high-lat) derived from our stratospheric CFC-11 measurements is consistent with our presented best estimate, $\epsilon_{app}$(CFC-11, high-lat) $= (-7.8 \pm 1.7)$ ‰. We do not reject our $\epsilon_{app}$(CFC-11, high-lat) derived directly from our CFC-11 observations as it is reasonable and derived from relatively many data. We used a

different estimate in our analysis simply to achieve the best possible model precision. In contrast, we do not believe our derived $\epsilon_{app}$(CFC-11, mid-lat) $= (-26.5 \pm 4.0)$ ‰ is correct based on consideration of previously reported $\epsilon_p$ (Zuiderweg et al., 2012, Table 2). Our $\epsilon_{app}$(CFC-11, mid-lat) derived from our stratospheric CFC-11 measurements is greater than previously reported $\epsilon_p$(CFC-11), which is inconsistent with our best understanding of atmospheric mixing (Kaiser et al., 2006). Also, our derived $\frac{\epsilon_{app}(\text{CFC-11, mid-lat})}{\epsilon_{app}(\text{CFC-12, mid-lat})} = 0.87 \pm 0.33$, which is inconsistent with $\frac{\epsilon_p(\text{CFC-11})}{\epsilon_p(\text{CFC-12})} = 0.39 \pm 0.02$. Our mid-latitude stratospheric CFC-11

regressions were derived from few data ($n = 5$) and are heavily influenced by one data point with $\ln(1-f) = -1.49$. We believe, with additional measurements, $\epsilon_{app}$(CFC-11) would likely decrease in magnitude.

*Author contributions.* MT prepared the manuscript with JCL, JK, and TR, with contributions from all authors. SA measured the firn air samples and AR measured the stratospheric samples, under the supervision of JCL, JK, and WTS. PM and EW provided the firn air modelling and JK provided the tropospheric modelling. TR was responsible for the Geophysika aircraft sampling while RM, TR, JK and WTS led the
firn sampling activities.

*Competing interests.* The authors declare no competing interests.

*Acknowledgements.* This work received funding from the European Research Council (EXC3ITE "EXploring Chemistry, Composition and Circulation in the stratosphere with Innovative TEchnologies", Grant Agreement 678904), the Horizon 2020 research and innovation programme through the EUROCHAMP-2020 Infrastructure Activity under Grant Agreement 730997, and the UK Natural Environment Research
Council (Research Fellowship NE/I021918/1). NEEM is directed and organised by the Centre for Ice and Climate at the Niels Bohr Institute and US NSF, Office of Polar Programs. It is supported by funding agencies and institutions in Belgium (FNRS-CFB and FWO), Canada (GSC), China (CAS), Denmark (FIST), France (IPEV, CNRS/INSU, CEA and ANR), Germany (AWI), Iceland (RannIs), Japan (NIPR), Korea (KOPRI), the Netherlands (NWO/ALW), Sweden (VR), Switzerland (SNF), United Kingdom (NERC) and the USA (US NSF, Office of Polar Programs). We thank Kelly Redeker for supplying raw data from Redeker et al. (2007). We also thank S. Montzka, J. W. Elkins,

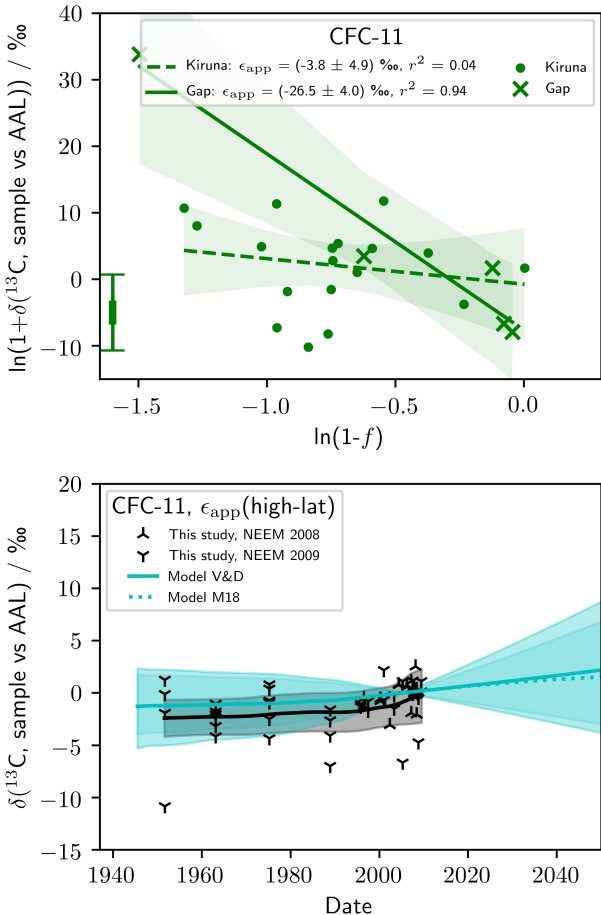

**Figure D1.** Rayleigh plot showing observations of $\delta^{13}$(C, CFC-11), and derived $\epsilon_{app}$(CFC-11, high-lat) and $\epsilon_{app}$(CFC-11, mid-lat) (top). Also shown is measured and modelled $\delta_T$(CFC-11) (see Figure 4 and Section 2.5 for a description of the model). The model was forced with $\epsilon_{app}$(CFC-11, high-lat). The errorbar in the bottom left corner of each graph shows the median repeatability of the reference gas measurements over the measurement days and the median error deriving from the fractional release factor.

and others involved in the NOAA-HATS program and NOAA/ESRL Global Monitoring Division for CFC-12 and CH₃Cl mole fraction data (ftp://aftp.cmdl.noaa.gov/data/hats/cfcs/cfc12/combined/ and ftp://aftp.cmdl.noaa.gov/data/hats/methylhalides/ch3cl/flasks/).

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
