# Peer review of "Stratospheric carbon isotope fractionation and tropospheric histories of CFC-11, CFC-12 and CFC-113 isotopologues"

_Atmospheric Chemistry and Physics, 2020_

## Referee Comment (RC1) · Anonymous Referee #1 · 14 Oct 2020

General comments:

In the submitted manuscript "Stratospheric carbon isotope fractionation and tropospheric histories of CFC-11, CFC-12 and CFC-113 isotopologues" Thomas et al. report on the first measurements of the stable carbon isotope composition of CFCs sampled in the stratosphere from which they derived isotopic enrichment factors (epsilon). In addition, firn air samples collected in 2008 and 2009 were re-measured in order to obtain the stable carbon isotope composition of CFCs in the troposphere. These data were used to model the change of delta13C in CFCs over time in the troposphere. Initially, this topic appears very interesting and I believe that the interpretation based

on the presented data is plausible. The major issue of this study is, however, the data itself. First of all, there is no information about the sampling procedures in this manuscript. This should be provided, at least in brief, in the appendix. It is not acceptable that the reader has to read several other papers in order to find information that is highly relevant to the current study. Furthermore, the used method, measurement of delta13C by GC-MS with a single detector, is completely new to me and I also did not find any published GC-MS method successfully demonstrating delta13C analysis at natural isotope abundance levels. Such methods need to be ground-truthed, that is, important parameters such as analytical precision, reproducibility, accuracy etc have to be evaluated and reported. The submitted manuscript contains no such information apart from a linearity check. It is also concerning that basic principles of stable isotope analysis are disregarded such as the use of several reference materials which are directly linked to the isotope-delta zero-point and realization of a two-point calibration in order to correct for scale contraction effects. This means, that the delta13C and epsilons are not comparable with other published values because the scale measured by this mass spec may differ from the official scale. Without a referencing procedure and two-point calibration, the uncertainty of the data can be considered substantially larger than presented (see also specific comments). A two-point calibration should be carried out in a sequence with the samples because the measured scale may even change from day to day. Therefore, unfortunately, a retrospective correction is not possible and the data would have to be re-measured using the appropriate reference materials and methods. Overall this paper presents an interesting research idea but all interpretation is based on very uncertain, not evaluable, data which cannot be compared to any other study. Therefore I cannot recommend this paper for publication. Further comments are given below:

—

Specific comments:

- Page 3 line 3-5: Strictly speaking, the Rayleigh model requires a first-order or pseudofirst order reaction. Two reactions (photolysis and O1D) and transport and mixing alto-gether would give an epsilon that will differ constantly depending on sampling height, temperature, mixing pattern (etc). Epsilon app is, for example, applied in microbiology to describe enrichment factors that are smaller due to a rate limitation. A constantly changing mixture of different processes will yield enrichment factors that are not repro-ducible. It will be difficult to quantify degradation rates with these kind of epsilons. How do the authors make sure that a specific sample is not just the result of mixing/dilution?

- Page 3 line 6-12: I'm not sure if these chlorine isotope measurements are of big help. Photolysis cleaves the C-Cl bond and therefore fractionation should occur at a similar rate for the isotopes of both C and Cl. It seems contradictory to me that there is no difference in fractionation between mid and high latitude samples for chlorine isotopes (Allin et al 2015) whereas for carbon a distinct difference is reported. One would expect that there is a latitudinal dependence of both C and Cl or no dependence for the both of them.

- Page 3 line 32-34: to be sure that the integration method in Zuiderweg et al (2013) works, one would have to show that different CFC-12 amounts/ peaksizes after the CH3Cl peak (which does not change much) would leave the CFC-12 signature un-changed. The baseline calculation used by Zuiderweg cuts away the front part of the peak and the smaller the CFC-12 peak, the more (relative to total peak area) is cut away. The frontpart is always heavier compared to the tail (e.g. Matucha et al 1991 Doi 10.1016/0021-9673(91)85030-J). This could be the reason for the very depleted values for firn air samples at 67m and 69m. This is partly also discussed in Appendix B but how the correction was carried out does not become clear. Please also define gamma(CH3Cl) and gamma (CFC-12) in Appendix B

- Page 4 line 15-18: What kind of MS is used? Stable carbon isotope measurements are usually carried out by isotope ratio mass spectrometers with several detectors (Faraday type) to allow for the simultaneous measurement of the masses. As far as I could find out, the tri-sector has only one detector which means switching between

masses and thus less precise measurements. I'm aware that stable chlorine isotopes can be measured in this way but precision is considerably worse compared to standard methods (DI/GC-IRMS, GC-MC-ICPMS). For stable carbon isotopes I did not find a published method for single detector MS being able to measure d13C at natural abundance and no information is given about the performance of this method (analytical precision, reproducibility, accuracy etc). A citation of the corresponding methods paper should be given and the most important parameters mentioned in the manuscript or much more information is required which could be given in the Appendix

- Page 5 line 5-8: Are these the only differences between method A and Method B? If the same instrument was used and only these few parameters were changed this brief description is sufficient. Calling it method A and B is confusing because the reader might think of different methods such as GC-IRMS, laser etc.

- Page 5 Line 9-11: It is quite concerning that only one standard was used on a regular MS system not being an isotope ratio mass spectrometer. The usual way would be to use three reference materials which were cross-calibrated against secondary (or at least tertiary reference materials) thus allowing to put the samples's isotopic values in relation to the 0-point of the scale (e.g. VPDB). Even if another zero-point is chosen, such as the mentioned air standard AAL, this two-point calibration procedure is necessary because the scales measured by each mass spec may be contracted or expanded. This means that, for example, 12 ‰ difference between two samples measured with one mass spec may be 10 or 13 with another. This effect of scale compression is relatively small for d13C measured with GC-IRMS but it can be quite large for GC-MS. For instance, Bernstein et al (2011, doi: 10.1021/ac200516c) showed that for chlorine the scales of different GCMS varied by plus/minus 30%. Since the abundance difference of the heavy (99% 12C) and the light carbon isotope (1% 13C) is much larger than for chlorine (76% for 35Cl, 24% for 37Cl) I would expect even larger uncertainties here and these uncertainties add to the already quite large analytical uncertainties shown in the paper.

- Page 5 Line 20: deriving the isotope ratio from the regression of the raw intensities is quite handy but from own experience I know that it does not work well for all methods. If the mass spec has only one detector (switching between the masses), the outcome is not a straight line but a hysteresis curve which produces a higher uncertainty than the usual integration approach (integrating the area under the peaks). There is also no information in the cited papers about the quality of this approach (e.g. R2 of the regression line).

- Figure 1: It is not clear to me, what the authors are correcting for. Transport is corrected in section 2.4 as far as I could understand. Also, given the spread of the d37Cl values, does this correction provide any improvement to the data?

- Page 7 line 12: Isn't the concentration of CFCs in firn air directly related to the "age"? Wouldn't that provide an independent tool to check modelling results? Or is it assumed that CFCs diffuse downward due to lower concentrations there? This would be a mixing problem again.

- Page 8 Line 3-10: As stated above, there is no certainty about the d13C scale because no cross calibration against international reference material was carried out. Re-measuring two samples does not give more certainty in this case because Zuiderweg et al do also not provide any details about two-point calibration, reference material etc. All data can only be treated as a rough approximation.

- Figure 3: Did the authors carry out a regression analysis? For instance for CFC-113 (Kiruna) the data is so scattered that I would assume they are not even correlated. Please provide R2 in the plots. Preferably also provide p-values of a statistical test or, if the authors prefer, use another measure of the effect size to show whether the data is correlated or not. There must also be something wrong with the confidence bounds given in the plots. 95% confidence interval means that it contains 95% of the data points (which they do not).

- Page 12 Line 31-32: How can values of about -60‰ (Zuiderweg) be consistent with

about -20‰ This is a comparison of apples and oranges. With an assumed scale factor eventually all data will be "consistent"

- Page 12 Line 33: I doubt that diffusion in the open atmosphere changes the isotopic composition in a way that would be relevant to this study. It is much slower than advection which does not cause fractionation.

- Page 13 Figure 4: Are the symbols at each time point indicating measurements of the same sample (replicates) or are they actually individual samples? Overall this comparison does not provide much information. The spread of the data is very large.

- Page 13 Line 2: Allin et al did not report a meridional difference. That was stated further above.

- Page 14 Table 4: There is more emission data out there in the literature. Phillips et al 2020 (DOI: 10.1021/acs.est.9b05746) reported Dual Inlet IRMS measurements of CFCs (and HCFCs) which are very precise and properly linked to the V-PDB scale

- Page 14 Line 6 This can only provide a very rough estimate because the errors for epsilon-CFC-12 are also scaled

- Page 14 Line 10-13 Does it mean that the modelling is based only on high latitude measurements taken above the polar circle (Kiruna)? These epsilons are smaller than those at mid latitudes. So the model would only make sense if one assumes that only in the high latitudes CFCs mix with the troposphere. Otherwise I would think that a weighted mean of the mid and high latitude epsilons should be calculated. This would still ignore low latitude fractionation for which no epsilons are known yet. Does the model account for mixing of stratospheric CFC (high and mid latitude) before they mix back into the troposphere? If not, would the model still fit the data if mid latitude epsilons are used? Maybe I missed it but this should be made clear.

—

Technical corrections:

- Page 1 Line 10: delta is expressed in an unusual way: $\delta$(13C). What is the rational of using parentheses? There are multiple good practice guides on how to properly report delta and epsilon (e.g. https://www.forensic-isotopes.org/gpg.html or Coplen 2011, DOI: 10.1002/rcm.5129)

- Page 3 line 15: please define epsilon p

- Page 3 line 17-18: The cause and effect relationship is mixed up here. It is not the values that lead to larger fractionation but the process (having shown large values in the laboratory).

- Page 5 line 16-17: Only every forth measurement was a reference. So samples are therefore not "bracketed" by reference measurements because this would require every second measurement to be a reference.

- Page 5 line 19 was AAL used as "bracketing standard"?

- Page 5 line 29: what is the meaning of temporal signal? The change of the isotopic signature over time? Please clarify here and further below.

- Page 5 line 26-30:It is not clear for what the correction is applied

- Page 8 line 26: Please define what the fractional release factor is. How is it calculated? Error bars for 1-f should be provided in Figure 3 and C1 (x-axis)

- Page 11 line 6: one could write "larger" because it is a larger isotope effect. The minus just means it is a normal isotope effect

- Page 11 line 9: "while epsilonapp(CFC-12) was most negative at high-latitudes" this is not consistent with table 2

- All Figures: It would be very helpful to see the error bars for each data point. If the same uncertainty is assumed for each sample the error bar can be presented as in Figure 3 ($\pm$ 6‰. Please also give the uncertainty for (1-f) and the calculated ages.

- Page 15 line 9: "caused by one measurement depth". What are the authors trying to say?

- Page 15-16 Conclusion section: This is just again a summary of the results. What are the implications of this study? Does it remove any uncertainty mentioned in the introductions?

---

## Short Comment (SC1) · 5 Nov 2020

We would like to provide an initial response to several concerns raised by reviewer #1, which in our view mispresent the findings of our manuscript. These concerns focus on methodological details, specifically the lack of standardisation against VPDB, lack of scale normalisation and the use of single-detector mass spectrometry. We conclude in the following that these concerns have no significant bearing on the results and none on the conclusions.

**1) Isotope delta standardisation**

We measure and report our $\delta(^{13}C)$ values against a reference tank containing dried tropospheric air (AAL-071170) at high pressure (collected at a northern hemisphere background site at Niwot Ridge, Colorado, USA, in summer 2005). From comparisons with similar tanks, we know that the CFC mole fractions and isotope ratios in this tank were stable over years, including the period of the measurements reported in the manuscript. Similarly, the samples (some of which were stored for more than 17 years before they were analysed by us) showed no significant long-term changes in their CFC mole fractions compared with measurements made nearer the time when they were collected.

A tropospheric air tank is an ideal reference material for our purposes because it is homogeneous, stable, widely available and comprises the same air matrix as the unknown sample. This tank (AAL-071170) defines the zero point of our isotope delta scale. The availability is not restricted to similar tanks of background air filled around the same time; actually, the troposphere as a whole can be used because of the long atmospheric lifetimes of the three CFC gases studied (52 to 100 years).

The focus of the manuscript is on relative variations in $\delta(^{13}C)$ over time (firn) and space (stratosphere) with respect to modern tropospheric air (chosen to be represented by the AAL-071170 tank). The detection and quantification of such changes do not require calibration against other reference materials (such as the virtual VPDB standard), which – as the reviewer correctly points out – would only lead to higher uncertainties in the reported $\delta(^{13}C)$ values.

The absence of a calibration against VDPB, or indeed the lack of SI traceability, is no impediment for the study of relative changes in gas or isotope ratios, as evidenced – for example – by atmospheric $O_2/N_2$ ratio measurements, which have been carried out and exploited successfully for more than 30 years' of carbon cycle research before an absolute calibration scale with an accuracy similar to the achievable measurement precision was developed (Aoki et al., 2019).

Similarly, variations in $N_2O$ isotopocule ratios in firn air and the stratosphere have been reported against uncalibrated in-house standards, without loss of relevance or credibility (Röckmann et al., 2003; Röckmann et al., 2001).

In particular, the apparent stratospheric isotope fractionations ($\varepsilon_{app}$) are entirely independent of the chosen isotope delta scale. Other than claimed in the review, they would therefore be easily comparable with other published stratospheric isotope fractionations, should additional measurements become available in the future. We are not aware of any measurements besides the ones we report.

Contrary to the reviewer's assertion, we can compare our $\delta(^{13}C)$ values with other measurements. In the manuscript, we have indeed compared our CFC-12 isotopologue ratio measurements in firn air (on the AAL-071170 scale) against analyses of the same samples using a GC-combustion-IRMS system, reported on the VPDB scale (Zuiderweg et al., 2013). This allowed determining the $\delta(^{13}C)$ value of CFC-12 in AAL-071170 on the VPDB scale as (–43.0±2.3) ‰ (Eq. 3 of the manuscript). A similar approach could be taken for CFC-11 and CFC-113 in AAL-071170 and at that time a retrospective correction be applied.

**2) Isotope scale normalisation**

The reviewer also criticised the lack of scale normalisation. Such scale normalisations are required where there is cross-contamination between samples, isotope exchange or blank effects (Kaiser, 2008), which generally lead to a delta scale contraction. Such corrections are usually of the order of <10 % of the delta differences. We cannot exclude the possibility that our method experiences scale contraction, but even a 10 % scale correction would be irrelevant, given the analytical precision we can achieve with our method. For example, the uncertainties in the firn air $\delta$ changes are between 30 and 60 % of the $\delta$ changes: (2.9±1.6) ‰ for CFC-11, (5.3±2.2) ‰ for CFC-12 and (9.3±2.7) ‰ for CFC-113 s (p. 8, l. 18). Having said that, we are confident that our analytical system does not suffer from memory effects, significant blanks or isotope exchange. The inlet is evacuated to < 0.1 mbar between runs and we have found no memory effects for our analytical species. All blank signals are well below 0.1 % of the reference tank peak area. Isotope exchange is unlikely to play a significant role due to the chemical inertness of the CFCs. This is reflected by their long-term stability in our tanks and canisters.

It is worth noting that the air volume of between 200 and 600 ml (20 ºC, 1bar) used to achieve this level of precision only yields 2 pmol for CFC-113, 12 pmol for CFC-12 and 6 pmol for CFC-11 at their modern tropospheric mole fractions. This low sample volume is a limitation imposed by the nature of the highly valuable firn and stratospheric samples. For comparison, the CFC amounts our method requires are a factor of $10^4$ to $10^5$ less than what Horst et al. (2015) have used to achieve a precision of 0.5 ‰ for $\delta(^{13}C)$. The CFC amounts stated here are for reference gas extractions; they are lower at stratospheric altitudes and the lower firn depths.

**3) Use of single-detector mass spectrometry**

Finally, the reviewer commented that single-detector mass spectrometry had not been used previously for carbon isotopes, but also acknowledged that it had found applications for chlorine (Laube et al., 2010; Aeppli et al., 2010; Allin et al., 2015) and bromine (Zakon et al., 2016; Horst et al., 2013) isotope systems. We would like to add sulfur (Angert et al., 2019), nitrogen and oxygen isotopes (Neubauer et al., 2020) to this list, as well as indeed carbon isotopes (Eiler et al., 2017; Hauri et al., 2002; Schutten et al., 1957; Nier, 1940). As the age of some of these references show, single-detector isotope ratio measurements are as old as the field of isotope geochemistry. Their main drawback is the reduced repeatability, but this may be more than offset by a reduced sample size requirement as in our case.

To further validate our single-detector method, we compared $\delta$ values of CFC-12 samples from laboratory photolysis experiments (Zuiderweg et al., 2012) against measurements of the same

samples on a GC-combustion IRMS system (Fig. 1). This required up to 1000-fold dilution of the photolysis samples to adjust their CFC-12 mole fractions of up to 530 nmol mol$^{-1}$ to match the CFC-12 mole fraction of 534 pmol mol$^{-1}$ in the AAL-071170 reference tank.

The regression coefficient of 1.00±0.09 shows good agreement between both methods, with a high correlation of $R^2$ = 0.92. This suggests that any scale contraction is likely to be small (or fortuitously similar for both methods). Either way, neither the choice of $\delta$ scale nor any scale contraction would change our conclusions regarding 1) changes in the tropospheric isotope delta, 2) the stratospheric isotope fractionation or, resulting from that, 3) the isotope deltas of the CFC emissions.

The *y*-axis offset of (46.3±2.7) ‰ can be converted to a $\delta$ value of (−44.2±2.5) ‰ for CFC-12 in AAL-071170 on the VDPB scale, which agrees within error with the value of (−43.0±2.3) ‰ obtained independently from the firn sample comparison (Eq. 3 of the manuscript).

In summary, we trust to have allayed the most pressing concerns of reviewer #1. We will provide a more detailed response after the end of the discussion phase.

[Figure]

Fig. 1: Comparison between $\delta$ values of CFC-12 samples from laboratory photolysis experiments measured using our single-detector method and measurements of the same samples on a GC-combustion IRMS system (Zuiderweg et al., 2012). The fit line was obtained by linear regression.

**References**

Aeppli, C., Holmstrand, H., Andersson, P., and Gustafsson, O.: Direct compound-specific stable chlorine isotope analysis of organic compounds with quadrupole GC/MS using standard isotope bracketing, Anal. Chem., 82, 420-426, 2010.

Allin, S. J., Laube, J. C., Witrant, E., Kaiser, J., McKenna, E., Dennis, P., Mulvaney, R., Capron, E., Martinerie, P., Röckmann, T., Blunier, T., Schwander, J., Fraser, P. J., Langenfelds, R. L., and Sturges, W. T.: Chlorine isotope composition in chlorofluorocarbons CFC-11, CFC-12 and CFC-113 in firn, stratospheric and tropospheric air, Atmos. Chem. Phys., 15, 6867-6877, 10.5194/acp-15-6867-2015, 2015.

Angert, A., Said-Ahmad, W., Davidson, C., and Amrani, A.: Sulfur isotopes ratio of atmospheric carbonyl sulfide constrains its sources, Scientific Reports, 9, 741, 10.1038/s41598-018-37131-3, 2019.

Aoki, N., Ishidoya, S., Matsumoto, N., Watanabe, T., Shimosaka, T., and Murayama, S.: Preparation of primary standard mixtures for atmospheric oxygen measurements with less than 1 $\mu$mol mol$^{-1}$ uncertainty for oxygen molar fractions, Atmos. Meas. Tech., 12, 2631-2646, 10.5194/amt-12-2631-2019, 2019.

Eiler, J., Cesar, J., Chimiak, L., Dallas, B., Grice, K., Griep-Raming, J., Juchelka, D., Kitchen, N., Lloyd, M., Makarov, A., Robins, R., and Schwieters, J.: Analysis of molecular isotopic structures at high precision and accuracy by Orbitrap mass spectrometry, Int. J. Mass Spectrom., 422, 126-142, 10.1016/j.ijms.2017.10.002, 2017.

Hauri, E. H., Wang, J., Pearson, D. G., and Bulanova, G. P.: Microanalysis of $\delta$13C, $\delta$15N, and N abundances in diamonds by secondary ion mass spectrometry, Chem. Geol., 185, 149-163, 10.1016/S0009-2541(01)00400-4, 2002.

Horst, A., Thornton, B. F., Holmstrand, H., Anderson, P., Crill, P. M., and Gustafsson, Ö.: Stable bromine isotopic composition of atmospheric $CH_3Br$, Tellus B, 65, 10.3402/tellusb.v65i0.21040, 2013.

Horst, A., Lacrampe-Couloume, G., and Sherwood Lollar, B.: Compound-specific stable carbon isotope analysis of chlorofluorocarbons in groundwater, Anal. Chem., 87, 10498-10504, 10.1021/acs.analchem.5b02701, 2015.

Kaiser, J.: Reformulated $^{17}O$ correction of mass spectrometric stable isotope measurements in carbon dioxide and a critical appraisal of historic `absolute' carbon and oxygen isotope ratios, Geochim. Cosmochim. Acta, 72, 1312-1334, 10.1016/j.gca.2007.12.011, 2008.

Laube, J. C., Kaiser, J., Sturges, W. T., Bönisch, H., and Engel, A.: Chlorine isotope fractionation in the stratosphere, Science, 329, 1167, 10.1126/science.1191809, 2010.

Neubauer, C., Crémière, A., Wang, X. T., Thiagarajan, N., Sessions, A. L., Adkins, J. F., Dalleska, N. F., Turchyn, A. V., Clegg, J. A., Moradian, A., Sweredoski, M. J., Garbis, S. D., and Eiler, J. M.: Stable isotope analysis of intact oxyanions using electrospray quadrupole-orbitrap mass spectrometry, Anal. Chem., 92, 3077-3085, 10.1021/acs.analchem.9b04486, 2020.

Nier, A. O.: A mass spectrometer for routine isotope abundance measurements, Rev. Sci. Instrum., 11, 212-216, 10.1063/1.1751688, 1940.

Röckmann, T., Kaiser, J., Brenninkmeijer, C. A. M., Crowley, J. N., Borchers, R., Brand, W. A., and Crutzen, P. J.: The isotopic enrichment of nitrous oxide ($^{15}N^{14}NO$, $^{14}N^{15}NO$,

$^{14}N^{14}N^{18}O$) in the stratosphere and in the laboratory, J. Geophys. Res., 106, 10403-10410, 10.1029/2000JD900822, 2001.

Röckmann, T., Kaiser, J., and Brenninkmeijer, C. A. M.: The isotopic fingerprint of the pre-industrial and the anthropogenic $N_2O$ source, Atmos. Chem. Phys., 3, 315-323, 10.5194/acp-3-315-2003, 2003.

Schutten, J., Boerboom, A. J. H., v. d. Hauw, T., and Monterie, F.: Precise measurement of isotope ratios with a single collector mass spectrometer, Applied Scientific Research, Section B, 6, 388-392, 10.1007/BF02920395, 1957.

Zakon, Y., Halicz, L., Lev, O., and Gelman, F.: Compound-specific bromine isotope ratio analysis using gas chromatography/quadrupole mass spectrometry, Rapid Commun. Mass Spectrom., 30, 1951-1956, 10.1002/rcm.7672, 2016.

Zuiderweg, A., Kaiser, J., Laube, J. C., Röckmann, T., and Holzinger, R.: Stable carbon isotope fractionation in the UV photolysis of CFC-11 and CFC-12, Atmos. Chem. Phys., 12, 4379-4385, 10.5194/acp-12-4379-2012, 2012.

Zuiderweg, A., Holzinger, R., Martinerie, P., Schneider, R., Kaiser, J., Witrant, E., Etheridge, D., Petrenko, V., Blunier, T., and Röckmann, T.: Extreme $^{13}C$ depletion of $CCl_2F_2$ in firn air samples from NEEM, Greenland, Atmos. Chem. Phys., 13, 599-609, 10.5194/acp-13-599-2013, 2013.

---

## Referee Comment (RC2) · Anonymous Referee #2 · 24 Nov 2020

This is clearly a very important subject, and an important, unique data set. It is therefore critical to make the most of the opportunity which I feel is not being done at present. More work should be performed in order to put the results of the analysis and modelling into context. 1. The reader is left themselves to try to make sense of the discrepancy in the delta\_T(13C, CFC-113) data before 1980. The authors write provocatively, 'While this discrepancy may be indicative of a change in d\_E(13C, CFC-113), it is premature to assign one.' I believe the discrepancy could also be indicative of faults with the sampling, analysis, and modelling and that either it is premature or it is not. The authors should decide if they have confidence in the conclusion and wish to defend it, or perhaps, based on statistics, model validation and so on, they would decide to withdraw.

If they support the conclusion I would suggest doing more work to investigate what this change would be, for example change in manufacturer or process. 2. More work is needed to put the results into perspective. What is known now that was not known before? How will the results be used? Were the CFC budgets under-constrained, or will these results provide additional insight into stratospheric changes or processes in firn, or ? 3. Does the derived stratospheric photolytic fractionation factor match the predictions of theory and experiment? 4. There are a number of technical issues listed by another reviewer which should be addressed.

---

## Author Comment (AC1)

Thanks to both reviewers for their time, their comments on this manuscript, and for expressing interest in the topic. We have already responded to several points made by Reviewer 1 in an author response (https://acp.copernicus.org/preprints/acp-2020-843/acp-2020-843-SC1-supplement.pdf) and here address their comments point by point, referring back to our author response where appropriate. We address the comments of Reviewer 2 for the first time.

Throughout, our response is in green, the reviewer comments are in black, deletions from our manuscript are in red, and insertions to our manuscript are in blue.

**Response to Reviewer 2**

R2 asked that we make fuller use of our data and analysis. We have addressed their specific points through changes to the text of the results, discussion, and conclusion of our manuscript.

R2 comment

1) More work should be performed in order to put the results of the analysis and modelling into context. 1. The reader is left themselves to try to make sense of the discrepancy in the delta_T(13C, CFC-113) data before 1980. The authors write provocatively, 'While this discrepancy may be indicative of a change in d_E(13C, CFC-113), it is premature to assign one.' I believe the discrepancy could also be indicative of faults with the sampling, analysis, and modelling and that either it is premature or it is not. The authors should decide if they have confidence in the conclusion and wish to defend it, or perhaps, based on statistics, model validation and so on, they would decide to withdraw. If they support the conclusion I would suggest doing more work to investigate what this change would be, for example change in manufacturer or process.

Author response

As stated in the manuscript, the discrepancy between our measurements and modelling for $\delta_T(^{13}C,$ CFC-113) highlights a possible change in $\delta_E(^{13}C,$ CFC-113). This discrepancy is based on measurements of one sample with a low mole fraction (mole fractions are now given in the supplement) and box modelling of $\delta_T(^{13}C,$ CFC-113). The discrepancy is only marginally significant, with the 95 % bounds on the measured and modelled trend a maximum of 1.9 ‰. We find no reason to exclude these data, having found no chromatographic interferences on the ions used, and no evidence for any artefacts from sampling or firn modelling as evidenced by the histories of multiple gases reconstructed from samples collected during those campaigns, including the chlorine isotopologues of the three CFCs presented here (e.g., Buizert et al., 2012; Witrant et al., 2012; Allin et al., 2015). However, we acknowledge that there are potential unknown unknowns, particularly at the low mole fraction of this sample. We stand by our conclusion: our analysis suggests a change in source signature but more work would be needed to definitively ascribe one.

According to Kirk-Othmer (1994) the two main feedstock materials for manufacturing CFC-113 have been hexachloroethane and tetrachloroethene. However, there are multiple processes that use CFC-113 as an intermediate or where it is a byproduct (Adcock et al., 2018), some of which were only introduced in the 1990s. In comparison to CFC-11 or CFC-12 there is therefore much greater scope for a change in isotopic signature of CFC-113 over the last decades. We compared

the change in $\delta_E(^{13}C, CFC\text{-}113)$ required to bring our modelled confidence bounds into line with the range of previously reported $\delta_E(^{13}C, CFC\text{-}113)$ – with a range of nearly 5 ‰ – in the discussion.

To address R2's concerns, we have:
changed the abstract

15  113) did not agree with our measurements earlier than 1980.  This discrepancy may be indicative of a change in

 $\delta_E(13C, CFC\text{-}113)$. However, this conclusion is based largely on a single sample and only just significant outside the 95 % confidence interval. Therefore more work is needed to independently verify this temporal trend in the global tropospheric $^{13}C$ isotopic composition of CFC-113. Our modelling predicts increasing $\delta_T(^{13}C, CFC\text{-}11)$,

changed the discussion

$\delta_E(^{13}C, CFC\text{-}11)$ since at least 1952. For CFC-113, our modelling did not agree with our measurements earlier than around 1980. This discrepancy may be indicative of a change in $\delta_E(^{13}C, CFC\text{-}113)$ though, given our measurements do not provide a complete picture of the fractionation in CFC-113 and given this discrepancy is caused by one measurement depth, these results

30  do not confirm a change in $\delta_E(^{13}C, CFC\text{-}113)$.  Multiple industrial processes use CFC-113 as a feedstock, or produce CFC-113 as an intermediate (Adcock et al., 2018), so a change in $\delta_E(^{13}C, CFC\text{-}113)$ is  plausible. The discrepancy between the modelled and reconstructed confidence bounds is at most -1.9 ‰. The range of published $\delta_E(CFC\text{-}113)$ is $(-31.3 \pm 0.5)$ ‰ to $(-26.5 \pm 0.8)$ ‰ (Table 4), around 5 ‰. The discrepancy seen for measured and modelled $\delta_T(CFC\text{-}113)$ can be accounted for by the range of published $\delta_E(CFC\text{-}113)$.

changed the conclusions

**5 Conclusions**

We have presented the first measurements of the $\delta(^{13}C)$ of CFC-11, -12, and -113 for stratospheric air samples, and derived
15  values for the apparent isotopic fractionation, $\epsilon_{app}$, at high- and mid-latitudes of: $\epsilon_{app}$(CFC-11, high-lat) $= (-7.8 + 1.7)$ ‰; $\epsilon_{app}$(CFC-11, mid-lat) $= (-11.7 \pm 4.2)$ ‰; $\epsilon_{app}$(CFC-12, high-lat) $= (-20.2 \pm 4.4)$ ‰; $\epsilon_{app}$(CFC-12, mid-lat) $= (-30.3 \pm 10.7)$ ‰; $\epsilon_{app}$(CFC-113, high-lat) $= (-9.4 \pm 4.4)$ ‰; and $\epsilon_{app}$(CFC-113, mid-lat) $= (-34.4 \pm 9.8)$ ‰. While for CFC-12 and -113 these estimates are independent, the $\epsilon_{app}$(CFC-11) estimates are not, having been derived by scaling our $\epsilon_{app}$(CFC-12) measurements. Further measurements of $\delta(^{13}C, CFC-11)$ in the stratosphere are required to estimate $\epsilon_{app}$(CFC-11) independent
20  of CFC-12. For CFC-113, these $\epsilon_{app}$ are only applicable to the CCl$_2$F fragment of the molecule. When used to model model tropospheric the tropospheric isotopic composition, $\delta_T(^{13}C)$, our derived $\epsilon_{app}$(high-lat) drive strong fractionation from the mid 1900s through to 2050. For CFC-12, modelled

We also reconstructed $\delta_T(^{13}C, CFC-12)$ was consistent with ) from firn air measurements. Comparing these with the model shows that the histories of $\delta_T(^{13}C, CFC-12)$ reconstructed from measurements of firn air when using a constant isotopic
25  composition of emissions, $\delta_E(^{13}CFC-11)$ and $\delta_T(^{13}C, CFC-12)$, for the entire period covered by measurements. Our results are therefore are consistent with a constant isotopic source composition, $\delta_E(^{13}C, CFC-12)$since around 1956 and are inconsistent with the extreme depletion in ), and with stratospheric processing as the sole sink of these chemicals. Our results contradict previous reports of extreme depletion for $\delta_T(^{13}C, CFC-12)$ and change in $\delta_E(^{13}C, CFC-12)$proposed by Zuiderweg et al. (2013) . Likewise, for CFC-11, our results are consistent with a constant $\delta_E(^{13}C, CFC-11)$ since 1952. For , Such extreme depletions
30  could have challenged the history of CFC-12 industrial processes and feedstocks; the current understanding of their atmospheric cycling; and/or raised questions about their inertness in the biogeosphere. The discrepancy between reconstructed and modelled $\delta_T(^{13}C, CFC-113$, our results are not consistent with a constant ) suggests a change in $\delta_E(^{13}C, CFC-113)$since 1975. While potentially indicative of a change in . Changes in industrial processes that produce CFC-113 – as an end product or byproduct – could explain such a discrepancy, and the range of reported $\delta_E(^{13}C, CFC-113)$ , this discrepancy is based on would be sufficient to cause such a discrepancy. We caution, however, that this discrepancy derives from only one sample and further

5  firn or tropospheric measurements are required to confirm this. Our modelling predicts a continuing takes into account the fractionation of only on CFC-113 fragment. Further work would be needed to definitively assign a change in $\delta_E(^{13}C, CFC-113)$. The modelled increase in $\delta_T(^{13}C)$ up to from 2009 through 2050 for each CFC. This increase is sensitive to new emissions, though . We compared future $\delta_T(^{13}C, CFC-11)$ trends in scenarios with/without new CFC-11 emissions. The difference between scenarios was within uncertainty bounds, showing better modelling precision and precise quantification of the iso-
10  topic composition of emissions would be needed to detect the isotopic signature of recently reported new CFC-11 emissions in background air.

2) More work is needed to put the results into perspective. What is known now that was not known before? How will the results be used? Were the CFC budgets under-constrained, or will these results provide additional insight into stratospheric changes or processes in firn, or ?

We have reconstructed tropospheric histories of the $^{13}C$ isotopologues of CFC-11, and CFC-12 over several decades and have shown that these are consistent with changes expected from stratospheric processing alone. This is important independent evidence as large isotopic changes (such as the one inferred by Zuiderweg et al. (2013)) would have challenged the current understanding of their atmospheric cycling and raised questions about their inertness in the biogeosphere. For $\delta_E(^{13}C, CFC-113)$, we find tentative evidence for significant changes before 1980, hinting at changes in production procedures and/or materials, though we cannot exclude other measurement artefacts. We also show the potential of our measurements to ascertain adherence to the Montreal Protocol, as the $^{13}C$ content of all three species should increase in the future – though, as we point out, better precision for $\epsilon_{app}$, and $\delta_E$ would be required to use this technique in background air.

Our stratospheric measurements provide the estimates of $\varepsilon_{app}$(CFC-11), $\varepsilon_{app}$(CFC-12), and $\varepsilon_{app}$(CFC-113). Our $\varepsilon_{app}$(CFC-12) measurement allows calculating $\delta_T$($^{13}C$, CFC-12), and hence to infer that $\delta_E$($^{13}C$, CFC-12) has probably not undergone a large change, unlike previously reported. Similarly, $\delta_E$($^{13}C$, CFC-11) is unlikely to have undergone a significant change, whereas our results hint at a possible change in $\delta_E$($^{13}C$, CFC-113). These are direct uses of our results, made clearer by adding:

We have made this clearer by revising the conclusions (please see full conclusions in response to previous comment).

3) Does the derived stratospheric photolytic fractionation factor match the predictions of theory and experiment?
For CFC-12, our measured apparent isotopic fractionation, $\varepsilon_{app}$, is in semi-quantitative agreement with previously reported photolytic fractionation, $\varepsilon_p$ (Zuiderweg et al. 2012). The agreement is semi-quantitative because atmospheric processes tend to reduce $\varepsilon_{app}$ by a factor of 2-3 relative to $\varepsilon_p$ (Kaiser et al. 2006), in the lower stratosphere. The $\varepsilon_p$(CFC-12) presented by Zuiderweg et al. (2012) is 2-3 times $\varepsilon_{app}$(CFC-12).

Our best estimate of $\varepsilon_{app}$(CFC-11) is based on the measurements of $\varepsilon_{app}$(CFC-12), rescaled using $\varepsilon_p$ values for CFC-11 and CFC-12 measured by Zuiderweg et al. (2012). We hence have no independent point of comparison for $\varepsilon_{app}$(CFC-11).

Our best estimate of $\varepsilon_{app}$(CFC-113) is, to the best of our knowledge, the only value present in the literature, and we are not aware of any measurements or theoretical predictions of $\varepsilon_p$(CFC-113).

Given the lack of points of comparison, we believe the text in the first paragraph of the discussion is sufficient.

4) There are a number of technical issues listed by another reviewer which should be addressed.
We refer R2 to our response to R1 (below).

**Response to Reviewer 1**

R1 had concerns about the quality of the data presented. These concerns were laid out in R1's preamble, and some were followed up with specific comments. We first respond to their preamble, referring often to our previous author comment dealing with these concerns (https://acp.copernicus.org/preprints/acp-2020-843/acp-2020-843-SC1-supplement.pdf). To translate the points made in the author comment to the manuscript we made many changes. The biggest and most important change is the addition of a new Appendix (see below), where we validate our methods.
New appendix: Appendix B: Comparison of GC-MS with GC-IRMS measurements

As a further check on the quality of our method, we have compared measurements made using GC-IRMS (Zuiderweg et al., 2011) of a suite of photolysis samples as presented in Zuiderweg et al. (2012) to our own measurements – using GC-MS – of those samples (Figure B1). These samples were not subject to the $CH_3Cl$ chromatographic interference seen in Zuiderweg et al. (2013)

5   . The samples were diluted by 1000 times before measurement on our system to accommodate the higher sensitivity of our GC-MS method.

The agreement between the methods is good (Figure B1). Linear regression gives a high regression coefficient ($r^2 = 0.92$) and a gradient consistent with unity ($1.0 + 0.1$). This agreement holds over a range of $\delta$ spanning almost 60 ‰. The intercept is ($46.3 + 2.7$) ‰. This intercept results in a $\delta$ value for CFC-12 in AAL-071170 on the VPDB scale of ($-44.2 + 2.5$) ‰ —

5 consistent with the ($43.0 \pm 2.3$) ‰ derived using Equation 3.

New appendix figure, showing good agreement between methods

[Figure]

**Figure B1.** Comparison of measurements presented by Zuiderweg et al. (2012) using GC-IRMS (Zuiderweg et al., 2011) to our own measurements of those same samples.

Response to preamble

Our overarching message when responding to the preamble is that the concerns raised have no significant bearing on the results we present, and no bearing on the conclusions we draw.

Preamble 1) there is no information about the sampling procedures in this manuscript. This should be provided, at least in brief, in the appendix. It is not accept- able that the reader has to read several other papers in order to find information that is highly relevant to the current study. Furthermore, the used method, measurement of delta13C by GC-MS with a single detector, is completely new to me and I also did not find any published GC-MS method successfully demonstrating delta13C analysis at natural isotope abundance levels. Such methods need to be ground truthed, that is, important parameters such as analytical precision, reproducibility, accuracy etc have to be evaluated and reported. The submitted manuscript contains no such information apart from a linearity check.

We have added the following to Section 2.1 to give the reader an overview of the sampling methods:

**2.1 Sample collection**

We present new data from two stratospheric and two tropospheric data sets (Table 1). One stratospheric data set, which we call 'Kiruna', was collected at  high-latitudes from high altitude Geophysica flights out of Kiruna, Sweden, using the BONBON-I, BONBON-II, and CLAIRE cryogenic whole air samplers (Laube et al., 2010b). The other, which we

15    call 'Gap', was collected at  mid-latitudes from balloons launched from Gap, France, using the whole air sampler of the then Max Planck Institute for Aeronomy (Kaiser et al., 2006). The firn air samples were collected at NEEM in northern Greenland during field campaigns in 2008 and 2009. Shallow ice cores were drilled, stopping every few meters for air sampling, with the borehole sealed off from ambient air using a bladder (Allin et al., 2015; ?).

There are several references using single detector GC-MS measuring natural abundance δ($^{13}$C) (please see also our author response):
Eiler et al., 2017; Hauri et al., 2002; Schutten et al., 1957; and Nier, 1940

For method A, an estimate of the precision of the methodology is given in Figures 3 and C1. We have added some explanation to the caption of Figure C1 so that it is consistent with Figure 3. In the original manuscript, the error bars for CFC-12 and CFC-11 were switched. We have amended that error in the plots. For method B we take the uncertainty from the loess regression of the firn profile to be our best estimate of the precision of our reconstructed firn profile. These are given in Section 3.1. Updated stratospheric plots are shown below.

[Figure]

**Figure 3.** Rayleigh plots of our stratospheric measurements. The linear regression (lines) and 95 % confidence bounds on the regression (shading) are shown for the high-latitude (Kiruna) and mid-latitude (Gap) data sets. The gradients of these regressions, corresponding to $\epsilon_{app}$, are given in the legend with one standard error. The errorbar in the bottom left corner of each graph shows the median repeatability of the reference gas measurements over the measurement days and the median error deriving from the fractional release factor. CFC-11 data are presented in Appendix D.

[Figure]

**Figure D1.** Rayleigh plot showing observations of $\delta^{13}$(C, CFC-11), and derived $\epsilon_{app}$(CFC-11, high-lat) and $\epsilon_{app}$(CFC-11, mid-lat) (top). Also shown is measured and modelled $\delta_T$(CFC-11) (see Figure 4 and Section 2.5 for a description of the model). The model was forced with $\epsilon_{app}$(CFC-11, high-lat). The errorbar in the bottom left corner of each graph shows the median repeatability of the reference gas measurements over the measurement days and the median error deriving from the fractional release factor.

Preamble 2) It is also concerning that basic principles of stable isotope analysis are disregarded such as the use of several reference materials which are directly linked to the isotope-delta zero-point and realization of a two-point calibration in order to correct for scale contraction effects. This means, that the delta13C and epsilons are not comparable with other published values because the scale measured by this mass spec may differ from the official scale. Without a referencing procedure and two-point calibration, the uncertainty of the data can be considered substantially larger than presented (see also specific comments). A two-point calibration should be carried out in a sequence with the samples because the measured scale may even change from day to day. Therefore, unfortunately, a retrospective correction is not possible and the data would have to be re-measured using the appropriate reference materials and methods

We provided an in depth response to this comment previously, which we reproduce here (https://acp.copernicus.org/preprints/acp-2020-843/acp-2020-843-SC1-supplement.pdf).

We measure and report our $\delta(^{13}C)$ values against a reference tank containing dried tropospheric air (AAL-071170) at high pressure (collected at a northern hemisphere background site at Niwot Ridge, Colorado, USA, in summer 2005). From comparisons with similar tanks, we know that the CFC mole fractions and isotope ratios in this tank were stable over years, including the period of the measurements reported in the manuscript. Similarly, the samples (some of which were stored for more than 17 years before they were analysed by us) showed no significant long-term changes in their CFC mole fractions compared with measurements made nearer the time when they were collected.

A tropospheric air tank is an ideal reference material for our purposes because it is homogeneous, stable, widely available and comprises the same air matrix as the unknown sample. This tank (AAL-071170) defines the zero point of our isotope delta scale. The availability is not restricted to similar tanks of background air filled around the same time; actually, the troposphere as a whole can be used because of the long atmospheric lifetimes of the three CFC gases studied (52 to 100 years).

The focus of the manuscript is on relative variations in $\delta(^{13}C)$ over time (firn) and space (stratosphere) with respect to modern tropospheric air (chosen to be represented by the AAL071170 tank). The detection and quantification of such changes do not require calibration against other reference materials (such as the virtual VPDB standard), which – as the reviewer correctly points out – would only lead to higher uncertainties in the reported $\delta(^{13}C)$ values.

The absence of a calibration against VDPB, or indeed the lack of SI traceability, is no impediment for the study of relative changes in gas or isotope ratios, as evidenced – for example – by atmospheric $O_2/N_2$ ratio measurements, which have been carried out and exploited successfully for more than 30 years' of carbon cycle research before an absolute calibration scale with an accuracy similar to the achievable measurement precision was developed (Aoki et al., 2019).

Similarly, variations in $N_2O$ isotopocule ratios in firn air and the stratosphere have been reported against uncalibrated in-house standards, without loss of relevance or credibility (Röckmann et al., 2003; Röckmann et al., 2001).

In particular, the apparent stratospheric isotope fractionations ($\varepsilon_{app}$) are entirely independent of the chosen isotope delta scale. Other than claimed in the review, they would therefore be easily comparable with other published stratospheric isotope fractionations, should additional measurements become available in the future. We are not aware of any measurements besides the ones we report.

Contrary to the reviewer's assertion, we can compare our $\delta(^{13}C)$ values with other measurements. In the manuscript, we have indeed compared our CFC-12 isotopologue ratio measurements in firn air (on the AAL-071170 scale) against analyses of the same samples using a GCcombustion-IRMS system, reported on the VPDB scale (Zuiderweg et al., 2013). This allowed determining the $\delta(^{13}C)$ value of CFC-12 in AAL-071170 on the VPDB scale as (–43.0±2.3) ‰ (Eq. 3 of the manuscript). A similar approach could be taken for CFC-11 and CFC-113 in AAL071170 and at that time a retrospective correction be applied.

The reviewer also criticised the lack of scale normalisation. Such scale normalisations are required where there is cross-contamination between samples, isotope exchange or blank effects (Kaiser, 2008), which generally lead to a delta scale contraction. Such corrections are usually of the order of <10 % of the delta differences. We cannot exclude the possibility that our method experiences scale contraction, but even a 10 % scale correction would be irrelevant, given the analytical precision we can achieve with our method. For example, the uncertainties in the firn air $\delta$ changes are between 30 and 60 % of the $\delta$ changes: (2.9±1.6) ‰ for CFC-11, (5.3±2.2) ‰ for CFC-12 and (9.3±2.7) ‰ for CFC-113. Having said that, we are confident that our analytical system does not suffer from memory effects, significant blanks or isotope exchange. The inlet is evacuated to < 0.1 mbar between runs and we have found no memory effects for our analytical species. All blank signals are well below 0.1 % of the reference tank peak area. Isotope exchange is unlikely to play a significant role due to the chemical inertness of the CFCs. This is reflected by their long-term stability in our tanks and canisters.

It is worth noting that the air volume of between 200 and 600 ml (20 ºC, 1bar) used to achieve this level of precision only yields 2 pmol for CFC-113, 12 pmol for CFC-12 and 6 pmol for CFC11 at their modern tropospheric mole fractions. This low sample volume is a limitation imposed by the nature of the highly valuable firn and stratospheric samples. For comparison, the CFC amounts our method requires are a factor of $10^4$ to $10^5$ less than what Horst et al. (2015) have used to achieve a precision of 0.5 ‰ for $\delta(^{13}C)$. The CFC amounts stated here are for reference gas extractions; they are lower at stratospheric altitudes and the lower firn depths.

We have checked our method against measurements made using GC-IRMS (Zuiderweg et al. 2012). The agreement is good over a range of 60 ‰, validating our methodology. We have added an appendix (B) showing this validation, and also a new figure B1 (please see start of response to R1).

Specific comments:
Page 3 line 3-5: Strictly speaking, the Rayleigh model requires a first-order or pseudo first order reaction. Two reactions (photolysis and O1D) and transport and mixing altogether would give an epsilon that will differ constantly depending on sampling height, temperature, mixing pattern (etc). Epsilon app is, for example, applied in microbiology to describe enrichment factors that are smaller due to a rate limitation. A constantly changing mixture of different processes will yield enrichment factors that are not reproducible. It will be difficult to quantify degradation rates with these kind of

epsilons. How do the authors make sure that a specific sample is not just the result of mixing/ dilution?

Rayleigh fractionation has been used as a model to define $\varepsilon_{app}$ from a range of stratospheric data sets (Kaiser et al., 2006). As stated in the paper, $\varepsilon_{app}$ is an empirical value, affected by photolysis, reaction with $O(^1D)$, plus transport, mixing, and dilution. $\varepsilon_{app}$ is therefore an appropriate value for box modelling the influence of stratosphere-troposphere exchange on the tropospheric isotope signature where these stratospheric processes are not individually resolved. The Rayleigh model is independent of the reaction order. It applies for any process, for which the relationship $dc(^{13}C)/dc(^{12}C) / [c(^{13}C) / c(^{12}C)] = 1 + \varepsilon = $ const. holds.

Page 3 line 6-12: I'm not sure if these chlorine isotope measurements are of big help. Photolysis cleaves the C-Cl bond and therefore fractionation should occur at a similar rate for the isotopes of both C and Cl. It seems contradictory to me that there is no difference in fractionation between mid and high latitude samples for chlorine isotopes (Allin et al 2015) whereas for carbon a distinct difference is reported. One would expect that there is a latitudinal dependence of both C and Cl or no dependence for the both of them.

We agree with the reviewer on this point. Based on what is known about compact tracer-tracer correlations in the lower stratosphere (e.g., Volk et al. 1997), we would not expect to find any significant latitude-dependence in the apparent stratospheric isotope fractionations for the range of fractional release factors our observations cover. As the reviewer mentions, Allin et al. (2015) did not find a significant latitude-dependence for $\varepsilon_{app}(^{37}Cl)$ in in CFC-11 and CFC-113. The differences seen for $\varepsilon_{app}(^{37}Cl)$ in CFC-12 at mid-latitudes and high-altitudes (Allin et al. 2015) are possibly down to statistical artefacts or, less likely, a decrease of $\varepsilon_{app}(^{37}Cl)$ with altitude. This is illustrated by re-analysing the high-latitude data of Allin et al. (2015), but restricting the analysis to a subset of the data for which $\ln(y/y_T) \geq -0.6$. This gives $\varepsilon_{app}(^{37}Cl) = (-9.1\pm1.4)$ ‰ instead of $(6.8\pm0.8)$ ‰ as reported by Allin et al. (2015). The value of $(-9.1\pm1.4)$ ‰ agrees, to within $2\sigma$, with the mid-latitude $\varepsilon_{app}(^{37}Cl)$ value of $(-12.2\pm1.6)$ ‰.

The lack of a latitude-dependence for lower-stratospheric Rayleigh fractionation is supported by stratospheric observations of other long-lived trace gases, in particular carbon and hydrogen isotope fractionation in $CH_4$ (Röckmann et al. 2011) and nitrogen and oxygen isotope fractionation in N2O (Kaiser et al. 2006), which are constrained by a much wider range of observations, with lower measurement uncertainties, than currently available for CFCs. $CH_4$ and $N_2O$ have global atmospheric mean lifetimes of 10 years and 123 years, respectively, which covers the range of lifetimes of CFC-11 (52 years), CFC-113 (93 years) and CFC-12 (102 years). The three CFCs also have the same chemical sinks as $N_2O$ – photolysis and oxidation by $O(^1D)$, in similar proportions as $N_2O$. We therefore do not expect these CFCs to behave any differently than $CH_4$ and $N_2O$.

We have removed the comparison between out measurements and Allin et al. from the results, and we have added the following text to the discussion.

uncertainties. Given our best understanding of compact tracer-tracer correlations in the lower stratosphere we do not expect

significant meridional differences in $\epsilon_{app}$ for the range of observed fractional releases ($\ln(1-f) > -0.6$) (Volk et al., 1997) . The lack of a latitude-dependence for lower-stratospheric Rayleigh fractionation is supported by stratospheric observations

of other long-lived trace gases, in particular carbon and hydrogen isotope fractionation in $CH_4$ (Röckmann et al., 2011) and nitrogen and oxygen isotope fractionation in $N_2O$ (Kaiser et al., 2006), which are constrained by a much wider range of observations, with lower measurement uncertainties, than currently available for CFCs. $CH_4$ and $N_2O$ have global atmospheric mean lifetimes of 10 years and 123 years, respectively, which covers the range of lifetimes of CFC-11 (52 years), CFC-113 (93 years) and CFC-12 (102 years). The three CFCs also have the same chemical sinks as $N_2O$ – photolysis and oxidation by $O(^1D)$, in similar proportions as $N_2O$. We therefore do not expect these CFCs to behave any differently than $CH_4$ and $N_2O$. The observed meridional differences could be statistical artefacts deriving from our poorly constrained $\epsilon_{app}$(mid-lat).

Page 3 line 32-34: to be sure that the integration method in Zuiderweg et al (2013) works, one would have to show that different CFC-12 amounts/ peaksizes after the CH3Cl peak (which does not change much) would leave the CFC-12 signature unchanged. The baseline calculation used by Zuiderweg cuts away the front part of the peak and the smaller the CFC-12 peak, the more (relative to total peak area) is cut away. The frontpart is always heavier compared to the tail (e.g. Matucha et al 1991 Doi 10.1016/0021-9673(91)85030-J). This could be the reason for the very depleted values for firn air samples at 67m and 69m. This is partly also discussed in Appendix B but how the correction was carried out does not become clear. Please also define gamma(CH3Cl) and gamma (CFC-12) in Appendix B

We have defined those gamma terms

We agree that this is a plausible mechanism for the artefact. We have added the following to clarify the correction performed

The cause of this discrepancy was likely a measurement artefact in Zuiderweg et al. (2013). In the method of Zuiderweg et al. (2013), methyl-chloride elutes before CFC-12, such that the tail of the . Zuiderweg et al. (2013) model the methyl-chloride peak must be modelled and accounted for in the tail using an exponentially decaying function, and subtract this signal from their CFC-12 peak before integration. Zuiderweg et al. (2013) performed a dilution series to evaluate their method, including

Page 4 line 15-18: What kind of MS is used? Stable carbon isotope measurements are usually carried out by isotope ratio mass spectrometers with several detectors (Faraday type) to allow for the simultaneous measurement of the masses. As far as I could find out, the tri-sector has only one detector which means switching between masses and thus less precise measurements. I'm aware that stable chlorine isotopes can be measured in this way but precision is considerably worse compared to standard methods (DI/GC-IRMS, GC-MC-ICPMS). For stable carbon isotopes I did not find a published method for single detector MS being able to measure d13C at natural abundance and no information is given about the performance of this method (analytical precision, reproducibility, accuracy etc). A citation of the corresponding methods paper should be given and

the most important parameters mentioned in the manuscript or much more information is required which could be given in the Appendix

Allin et al. (2015) detail the bulk of the methodology used in this paper. All changes to their method are detailed in Section 2.3 and 2.3. There is no difference in the fundamental principles of the method for carbon and chlorine isotopes, other than natural abundance ratios being a factor of 29 lower for $^{13}C/^{12}C$ than for $^{37}Cl/^{35}Cl$, but the resulting loss on signal-to-noise ratio is partly offset by the relative isotope effects being larger for $^{13}C/^{12}C$.

For additional references of single-detector carbon isotope mass spectrometry, see Eiler et al., 2017; Hauri et al., 2002; Schutten et al., 1957; and Nier, 1940

Please see the additional appendix (B) with method validation.

Page 5 line 5-8: Are these the only differences between method A and Method B? If the same instrument was used and only these few parameters were changed this brief description is sufficient. Calling it method A and B is confusing because the reader might think of different methods such as GC-IRMS, laser etc.

Yes, those were the only differences.

We have added some clarification that each method uses the same instrument.

10   instrument settings to Allin et al. (2015).  Method B uses the same MS and chromatography, but we increased the detector voltage (from 375 to 400 V), reduced the number of mass fragments measured at any given time, and optimised our source and collector slit parameters for maximum signal.

Page 5 Line 9-11: It is quite concerning that only one standard was used on a regular MS system not being an isotope ratio mass spectrometer. The usual way would be to use three reference materials which were cross-calibrated against secondary (or at least tertiary reference materials) thus allowing to put the samples's isotopic values in relation to the 0-point of the scale (e.g. VPDB). Even if another zero-point is chosen, such as the mentioned air standard AAL, this two-point calibration procedure is necessary because the scales measured by each mass spec may be contracted or expanded. This means that, for example, 12 ‰ difference between two samples measured with one mass spec may be 10 or 13 with another. This effect of scale compression is relatively small for d13C measured with GC-IRMS but it can be quite large for GC-MS. For instance, Bernstein et al (2011, doi: 10.1021/ac200516c) showed that for chlorine the scales of different GCMS varied by plus/minus 30%. Since the abundance difference of the heavy (99% 12C) and the light carbon isotope (1% 13C) is much larger than for chlorine (76% for 35Cl, 24% for 37Cl) I would expect even larger uncertainties here and these uncertainties add to the already quite large analytical uncertainties shown in the paper.

We refer back to our previous response to R1, and our response earlier in this document to a similar comment in the preamble. Scale normalisations are required where there is cross-contamination between samples, isotope exchange or blank effects (Kaiser, 2008), which generally lead to a delta scale contraction. Such corrections are usually of the order of <10 % of the delta differences. We cannot exclude the possibility that our method experiences scale contraction, but even a 10 % scale correction would be irrelevant, given the analytical precision we can achieve with our method. For example, the uncertainties in the firn air δ changes are between 30 and 60 % of the δ changes: (2.9±1.6) ‰ for CFC-11, (5.3±2.2) ‰ for CFC-12 and (9.3±2.7) ‰ for CFC-113 s (p. 8, l. 18). Having said that, we are confident that our analytical system does not suffer from memory effects, significant blanks or isotope exchange. The inlet is evacuated to < 0.1 mbar

between runs and we have found no memory effects for our analytical species. All blank signals are well below 0.1 % of the reference tank peak area. Isotope exchange is unlikely to play a significant role due to the chemical inertness of the CFCs. This is reflected by their long-term stability in our tanks and canisters.

It is worth noting that the air volume of between 200 and 600 ml (20 ºC, 1bar) used to achieve this level of precision only yields 2 pmol for CFC-113, 12 pmol for CFC-12 and 6 pmol for CFC11 at their modern tropospheric mole fractions. This low sample volume is a limitation imposed by the nature of the highly valuable firn and stratospheric samples. For comparison, the CFC amounts our method requires are a factor of 104 to 105 less than what Horst et al. (2015) have used to achieve a precision of 0.5 ‰ for δ( 13 C). The CFC amounts stated here are for reference gas extractions; they are lower at stratospheric altitudes and the lower firn depths.

Our method has been validated over a wide range of δ values by comparing to measurements of the same samples made by Zuiderweg et al. (2012), showing scale effects have little effect on our results and none one our conclusions. Please see additional appendix (B) with method validation.

Page 5 Line 20: deriving the isotope ratio from the regression of the raw intensities is quite handy but from own experience I know that it does not work well for all methods. If the mass spec has only one detector (switching between the masses), the outcome is not a straight line but a hysteresis curve which produces a higher uncertainty than the usual integration approach (integrating the area under the peaks). There is also no information in the cited papers about the quality of this approach (e.g. R2 of the regression line).
We used the regression method because it gave better precision than the peak area method (Allin, 2015).

- Figure 1: It is not clear to me, what the authors are correcting for. Transport is corrected in section 2.4 as far as I could understand. Also, given the spread of the d37Cl values, does this correction provide any improvement to the data?
This correction allows us to use measurements of R(102/105) rather than R(102/101) for method B samples. By predicting R(105/101) using the firn model, we can use Equation 2 to recover R(102/101), which is our desired ratio. The correction is small and within the uncertainty bounds on our firn reconstructions shallow in the firn. Deeper in the firn, the correction reached around 4 ‰, which is significant.

We have clarified the purpose of this correction by adding

1. Substituting Equation 2 into 1 allows us to recover $\delta$ values from measurements of $R(102/105)$. With this treatment we use $^{12}C^{37}Cl_2F_2$ as a standard, assuming  $\delta_L(^{37,37}Cl)$ for CFC-11, -12, and  -113 is determined dominantly by diffusive and gravitational fractionation in the firn. As a check on our correction, we

- Page 7 line 12: Isn't the concentration of CFCs in firn air directly related to the "age"? Wouldn't that provide an independent tool to check modelling results? Or is it assumed that CFCs diffuse downward due to lower concentrations there? This would be a mixing problem again.
The concentration of CFCs, and the isotopologues of them, is related to their age in the firn, plus gravitational and diffusive mixing, which is what the model calculates. This model has been

previously demonstrated to work well for multiple trace gases with different concentration gradients and physico-chemical properties (Buizert et al. 2012, Witrant et al. 2012).

- Page 8 Line 3-10: As stated above, there is no certainty about the d13C scale because no cross calibration against international reference material was carried out. Re-measuring two samples does not give more certainty in this case because Zuiderweg et al do also not provide any details about two-point calibration, reference material etc. All data can only be treated as a rough approximation.
We refer back to our previous response to R1, and our response earlier in this document to a similar comment in the preamble.

- Figure 3: Did the authors carry out a regression analysis? For instance for CFC-113 (Kiruna) the data is so scattered that I would assume they are not even correlated. Please provide R2 in the plots. Preferably also provide p-values of a statistical test or, if the authors prefer, use another measure of the effect size to show whether the data is correlated or not. There must also be something wrong with the confidence bounds given in the plots. 95% confidence interval means that it contains 95% of the data points (which they do not).
We have added $r^2$ to the legends of the stratospheric plots (Figures 3 and D1).

We have added p values for the gradient of each of the regressions. Note that, for CFC-12 and -113, each gradient and hence $\varepsilon_{app}$, is significant with at least 90 % confidence. Each high latitude $\varepsilon_{app}$ is significant at 95 % confidence. We acknowledge that the mid-latitude $\varepsilon_{app}$ are poorly constrained and only use our high latitude $\varepsilon_{app}$ values for the stratosphere-troposphere box model calculations.

15      From our stratospheric measurements, we derived $\varepsilon_{app}$(CFC-12, high-lat) $= (-20.2 + 4.4)$ ‰ $(p < 0.01)$, $\varepsilon_{app}$(CFC-12, mid-lat) $= (-30.3 \pm 10.7)$ ‰ $(p = 0.07)$, $\varepsilon_{app}$(CFC-113, high-lat) $= (-9.4 \pm 4.4)$ ‰ $(p = 0.04)$, and $\varepsilon_{app}$(CFC-113, mid-lat) $= (-34.4 \pm 9.8)$ ‰ ($p = 0.04$)  Table 2. Of these, $\varepsilon_{app}$(CFC-12, mid-lat) is significant at 90 % confidence, with the others significant at 95 %.

The confidence bounds give the 95 % confidence interval for the regression model. These are different to the 95 % prediction intervals, which would encompass 95 % of the data. Here is Figure 3 with 95 % prediction intervals, which are much larger than the 95 % confidence bounds.

[Figure]

- Page 12 Line 31-32: How can values of about -60‰ (Zuiderweg) be consistent with about -20‰ This is a comparison of apples and oranges. With an assumed scale factor eventually all data will be "consistent"

In this case the scale factor is based on previous comparisons of photolytic and apparent fractionation for δ($^{15}$N, N$_2$O) and δ($^{18}$O, N$_2$O) reported in Kaiser et al. (2006). This scale factor is roughly 2 to 3, and hence our values of around -20‰ are consistent with a range from -40 to -60 ‰. This is not a quantitative comparison but is the only point of comparison available.

This comparison is particularly useful for CFC-11, where it allowed us to identify our ε$_{app}$($^{13}$C, CFC-11) as biased high. In this case, ε$_{app}$($^{13}$C, CFC-11, mid-lat) ≈ ε$_p$($^{13}$C, CFC-11) which is inconsistent with previously reported scale factors. Combined with the small sample size (n=5), this comparison highlighted our calculated ε$_{app}$($^{13}$C, CFC-11, mid-lat) as spurious.

- Page 12 Line 33: I doubt that diffusion in the open atmosphere changes the isotopic composition in a way that would be relevant to this study. It is much slower than advection which does not cause fractionation.

In fact, diffusion is responsible for the attenuation of the intrinsic photochemical fractionation to the observed apparent isotope fractionation, which is a factor of 2 to 3 times lower (Kaiser et al. 2006).

- Page 13 Figure 4: Are the symbols at each time point indicating measurements of the same sample (replicates) or are they actually individual samples? Overall this comparison does not provide much information. The spread of the data is very large.

The individual points show replicates. This figure shows that our reconstructed trends are consistent with our model (CFC-11 and CFC-12), that there is a discrepancy between our reconstructed CFC-113 trend and our model, and that the reconstructed trend of Zuiderweg et al. (2013) is inconsistent with our reconstructed trend and our modelling. The uncertainties, though admittedly rather large, are sufficient to draw these conclusions.

- Page 13 Line 2: Allin et al did not report a meridional difference. That was stated further above.

In fact, Allin et al. did report a meridional difference for CFC-12. We do not expect to find significant meridional differences in $\varepsilon_{app}$ over the range of fractional release factors covered by our mid-latitude $\varepsilon_{app}$. We have removed this comparison in the results.

negative at mid-latitudes, which is qualitatively consistent with previous measurements of $\epsilon_{app}(^{37}Cl)$ for these chemicals (Allin et al., 2015; Laube et al., 2010a). For each CFC, high-latitude . High-latitude $\epsilon_{app}$ were derived from more data than

- Page 14 Table 4: There is more emission data out there in the literature. Phillips et al 2020 (DOI: 10.1021/acs.est.9b05746) reported Dual Inlet IRMS measurements of CFCs (and HCFCs) which are very precise and properly linked to the V-PDB scale

Thanks for this useful reference. We've added the compositions from Phillips et al. (2019) and Horst et al. (2015) that were missing to Table 4

and have changed the text of Section 3.3

Table 4. $\delta_E(^{13}C)$ as predicted by our modelling and as reported in previous studies for CFC-11, -12, and -113. With the exception of Ertl (1997), all uncertainties are two standard errors.

| | $\delta(^{13}C,$ sample vs AAL) / ‰ This study | $\delta(^{13}C,$ sample vs VPDB) / ‰ | | | | | |
| --- | --- | --- | --- | --- | --- | --- | --- |
| | | This study | Ertl (1997) | Thompson et al. (2002) | Archbold et al. (2005) | Horst et al. (2015) | Phillips et al. (2019) |
| CFC-11 | $-2.7 \pm 0.5$ | | $-35$ to $-25$ | | $-26.2 \pm 0.6$ | $-33.34 \pm 0.07$ $-28.93 \pm 0.04$ | $-33.36 \pm 0.04$ |
| CFC-12 | $-4.3 \pm 1.3$ | $-47.1 \pm 1.3$ | $-45$ to $-33$ | | $-46.8 \pm 0.2$ | | $-45.27 \pm 0.04$ |
| CFC-113 | $-1.7 \pm 1.1$ | | | $-31.3 \pm 0.5$ | $-26.5 \pm 0.8$ | $-28.07 \pm 0.05$ | $-29.93 \pm 0.06$ |

reported $\delta(^{13}C,$ sample vs VPDB) values of CFC-12 gas that was purchased from manufacturers. Ertl (1997), as reported in Archbold et al. (2012), measured the $\delta(^{13}C)$ of gases sourced from several manufacturers, reporting a range of -45 ‰ to -33 ‰. Archbold et al. (2005) reported the $\delta(^{13}C)$ of three CFC-12 standards as , which range from $(-33)$ ‰ (Ertl, 1997) to $(-46.8 + 0.2)$ ‰ (Archbold et al., 2012) (Table 4). We modelled $\delta_E(^{13}C, CFC-12) = (-47.1 + 1.3)$ ‰ (two standard errors),
10  within the range of previously reported $\delta_E(^{13}C, CFC-12)$.

- Page 14 Line 6 This can only provide a very rough estimate because the errors for epsilon-CFC-12 are also scaled

The uncertainties on $\varepsilon_{app}(CFC-11)$ were calculated by propagating the errors on our measurements of $\varepsilon_{app}(CFC-12)$ and the errors on $\varepsilon_p(CFC-11)$ and $\varepsilon_p(CFC-12)$ as measured by Zuiderweg et al. (2012). We disagree that this is a rough estimate – rather, it is an estimate with quantified uncertainty.

- Page 14 Line 10-13 Does it mean that the modelling is based only on high latitude measurements taken above the polar circle (Kiruna)? These epsilons are smaller than those at mid latitudes. So the model would only make sense if one assumes that only in the high latitudes CFCs mix with the troposphere. Otherwise I would think that a weighted mean of the mid and high latitude epsilons should be calculated. This would still ignore low latitude fractionation for which no epsilons are known yet. Does the model account for mixing of stratospheric CFC (high and mid latitude) before they mix back into the troposphere? If not, would the model still fit the data if mid latitude epsilons are used? Maybe I missed it but this should be made clear.

From our understanding of compact tracer-tracer relationships in the lower stratosphere in general and Rayleigh-type fractionation of long-lived trace gases and their isotopologues in particular, we do not expect to see latitude-dependent differences in $\varepsilon_{app}$. See also our reply to another comment above. We have therefore used the statistically best-constrained high-latitude $\varepsilon_{app}$ values because they were derived from more data and we therefore have more confidence in them. The mid-latitude $\varepsilon_{app}$ are poorly constrained, and we therefore have less confidence in them and modelling resulting from them.

R1 also had some technical corrections:

C6- Page 1 Line 10: delta is expressed in an unusual way: $\delta$(13C). What is the rational of using parentheses? There are multiple good practice guides on how to properly report delta and epsilon (e.g. https://www.forensic-isotopes.org/gpg.html or Coplen 2011, DOI: 10.1002/rcm.5129)

This notation follows long-standing international conventions on the notation of physical quantity symbols and any associated labels, see, for example, the recommendations in the IUPAC Green Book (https://iupac.org/what-we-do/books/greenbook). Coplen (2011) recognizes the correct notation in a footnote, but expressed a personal preference for the incongruent notation without parentheses.

- Page 3 line 15: please define epsilon p
Done

- Page 3 line 17-18: The cause and effect relationship is mixed up here. It is not the values that lead to larger fractionation but the process (having shown large values in the laboratory).
We have reworded:

‰ at 203 K to $(-23.0 \pm 1.1)$ ‰ at 233 K) and CFC-12 ($(-66.2 \pm 3.1)$ ‰ at 203 K to $(-55.3 \pm 3.0)$ ‰ at 233 K). These values

20   imply greater levels of fractionation for $\delta(^{13}C)$ than for $\delta(^{37}Cl)$ in the stratosphere.

- Page 5 line 16-17: Only every forth measurement was a reference. So samples are therefore not "bracketed" by reference measurements because this would require every second measurement to be a reference.
We have reworded:

20  $100.9 \approx 101$) ion fragments for the sample and the  reference gas, respectively. $R_{std}$ was taken to be the weighted mean ratio of two  preceding and subsequent reference gases. Measured $\delta(^{13}C, CFC-11)$ and $\delta(^{13}C, CFC-12)$

- Page 5 line 19 was AAL used as "bracketing standard"?

We have replaced 'standard' with 'reference gas', which we believe clears this up (please see response to previous comment).

- Page 5 line 29: what is the meaning of temporal signal? The change of the isotopic signature over time? Please clarify here and further below.

We have clarified:

1. Substituting Equation 2 into 1 allows us to recover $\delta$ values from measurements of $R(102/105)$. With this treatment we use

5    $^{12}C^{37}Cl_2F_2$ as a standard, assuming $\delta_t(^{37,37}Cl)$ for CFC-11, -12, and

 -113 is determined dominantly by diffusive and gravitational fractionation in the firn. As a check on our correction, we

- Page 5 line 26-30: It is not clear for what the correction is applied

We have clarified (please see response to previous comment).

- Page 8 line 26: Please define what the fractional release factor is. How is it calculated? Error bars for 1-f should be provided in Figure 3 and C1 (x-axis)

We have clarified. Please note, the reference gives the method of calculation.

[Figure]

**Figure 3.** Rayleigh plots of our stratospheric measurements. The linear regression (lines) and 95 % confidence bounds on the regression (shading) are shown for the high-latitude (Kiruna) and mid-latitude (Gap) data sets. The gradients of these regressions, corresponding to $\epsilon_{app}$, are given in the legend with one standard error. The errorbar in the bottom left corner of each graph shows the median repeatability of the reference gas measurements over the measurement days and the median error deriving from the fractional release factor. CFC-11 data are presented in Appendix D.

A horizontal errorbar has been added to Figures 3 and D1 to show the error in ln(1-f). This error is small (1-2 %).

[Figure]

**Figure D1.** Rayleigh plot showing observations of $\delta^{13}$(C, CFC-11), and derived $\epsilon_{app}$(CFC-11, high-lat) and $\epsilon_{app}$(CFC-11, mid-lat) (top). Also shown is measured and modelled $\delta_T$(CFC-11) (see Figure 4 and Section 2.5 for a description of the model). The model was forced with $\epsilon_{app}$(CFC-11, high-lat). The errorbar in the bottom left corner of each graph shows the median repeatability of the reference gas measurements over the measurement days and the median error deriving from the fractional release factor.

Our stratospheric measurements are presented as Rayleigh plots in Figure 3, where $f$ is the fractional release factor quantifying the degree of stratospheric destruction (Leedham Elvidge et al., 2018). Destruction of CFC-12 and -113 (corresponding to an

- Page 11 line 6: one could write "larger" because it is a larger isotope effect. The minus just means it is a normal isotope effect
We have stuck with more negative as we feel this terminology is totally unambiguous.

- Page 11 line 9: "while epsilonapp(CFC-12) was most negative at high-latitudes" this is not consistent with table 2
This statement is consistent with table 2. Note that we are comparing the different CFCs a high-latitudes, not CFC-12 at high-latitude with CFC-12 at mid-latitude, which we have clarified by adding:

| CFC | $\epsilon_{app}$ /‰ | |
|-----|---------------|------------|
| | High-latitude | Mid-latitude |
| 11 | $-7.8 \pm 1.7^{\dagger}$ | $-11.7 \pm 4.2^{\dagger}$ |
| 12 | $-20.2 \pm 4.4$ | $-30.3 \pm 10.7$ |
| 113 | $-9.4 \pm 4.4$ | $-34.4 \pm 9.8$ |

(Allin et al., 2015; Laube et al., 2010a). For each CFC, high-latitude . High-latitude $\epsilon_{app}$ were derived from more data than

5 the mid-latitude $\epsilon_{app}$. Of the three CFCs, $\epsilon_{app}$(CFC-11) was least negative at both latitudes, while $\epsilon_{app}$(CFC-12) was most negative at high-latitudes and $\epsilon_{app}$(CFC-113) was most negative at mid-latitudes. We took $\epsilon_{app}$(high-lat) forward for our modelling because these were derived from more data and we have more confidence in them.

- All Figures: It would be very helpful to see the error bars for each data point. If the same uncertainty is assumed for each sample the error bar can be presented as in Figure 3 (± 6‰. Please also give the uncertainty for (1-f) and the calculated ages.
We have added the 1-f uncertainty to Figures 3 and C1 (see above). This uncertainty encompasses, among other important uncertainties, the error on the calculated ages.
For the Figures 2 and 4 we feel the error on the loess regression is the best estimate of the uncertainty and have retained these figures as is.

- Page 15 line 9: "caused by one measurement depth". What are the authors trying to say?
We have done some more work on the CFC-113 discrepancy in response to R2 (please see above). Please see the additional qualifications in the abstract and discussion.

- Page 15-16 Conclusion section: This is just again a summary of the results. What are the implications of this study? Does it remove any uncertainty mentioned in the introductions?
We have revised the conclusions in response to both reviewers, please see below.

**5 Conclusions**

We have presented the first measurements of the $\delta(^{13}C)$ of CFC-11, -12, and -113 for stratospheric air samples, and derived
15    values for the apparent isotopic fractionation, $\epsilon_{app}$, at high- and mid-latitudes of: $\epsilon_{app}$(CFC-11, high-lat) $= (-7.8 + 1.7)$ ‰;
$\epsilon_{app}$(CFC-11, mid-lat) $= (-11.7 \pm 4.2)$ ‰; $\epsilon_{app}$(CFC-12, high-lat) $= (-20.2 \pm 4.4)$ ‰; $\epsilon_{app}$(CFC-12, mid-lat) $= (-30.3 \pm 10.7)$
‰; $\epsilon_{app}$(CFC-113, high-lat) $= (-9.4 \pm 4.4)$ ‰; and $\epsilon_{app}$(CFC-113, mid-lat) $= (-34.4 \pm 9.8)$ ‰. While for CFC-12 and -
113 these estimates are independent, the $\epsilon_{app}$(CFC-11) estimates are not, having been derived by scaling our $\epsilon_{app}$(CFC-12)
measurements. Further measurements of $\delta(^{13}C,$ CFC-11) in the stratosphere are required to estimate $\epsilon_{app}$(CFC-11) independent
20    of CFC-12. For CFC-113, these $\epsilon_{app}$ are only applicable to the CCl$_2$F fragment of the molecule. When used  model
 the tropospheric isotopic composition, $\delta_T(^{13}C)$, our derived $\epsilon_{app}$(high-lat) drive strong fractionation from the mid
1900s through to 2050. For CFC-12, modelled

   We also reconstructed $\delta_T(^{13}C,$  ) from firn air measurements. Comparing these with the model
shows that the histories of $\delta_T(^{13}C,$
25    $\delta_E(^{13}C,$CFC-11) and $\delta_T(^{13}C,$ CFC-12)
 are consistent with a constant isotopic source composition, $\delta_E(^{13}C,$
 ), and with stratospheric processing as the sole sink of these chemicals. Our results contradict
previous reports of extreme depletion for $\delta_T(^{13}C,$ CFC-12) and  $\delta_E(^{13}C,$ CFC-12)
. Likewise, for CFC-11, our results are consistent with a constant $\delta_E(^{13}C,$ CFC-11) since 1952. For Such extreme depletions
30   could have challenged the history of CFC-12 industrial processes and feedstocks; the current understanding of their atmospheric
cycling; and/or raised questions about their inertness in the biogeosphere. The discrepancy between reconstructed and modelled
$\delta_T(^{13}C,$ CFC-113) suggests a change in $\delta_E(^{13}C,$ CFC-113)
. Changes in industrial processes that produce CFC-113 – as an end product or byproduct
– could explain such a discrepancy, and the range of reported $\delta_E(^{13}C,$ CFC-113)  would be
sufficient to cause such a discrepancy. We caution, however, that this discrepancy derives from only one sample and

5    takes into account the
fractionation of only on CFC-113 fragment. Further work would be needed to definitively assign a change in $\delta_E(^{13}C,$ CFC-113).
The modelled increase in $\delta_T(^{13}C)$  from 2009 through 2050  is sensitive to new emis-
sions. We compared future $\delta_T(^{13}C,$ CFC-11) trends in scenarios with/without new CFC-11 emissions. The difference
between scenarios was within uncertainty bounds, showing better modelling precision and precise quantification of the iso-
10   topic composition of emissions would be needed to detect the isotopic signature of recently reported new CFC-11 emissions
in background air.

**References**

Allin, S.: Trace gases in Antarctica and Greenland firn and ice: a record of carbonyl sulphide and the isotopologues of chlorofluorocarbons, PhD thesis, University of East Anglia, 2015

Allin, S. J., Laube, J. C., Witrant, E., Kaiser, J., McKenna, E., Dennis, P., Mulvaney, R., Capron, E., Martinerie, P., Röckmann, T., Blunier, T., Schwander, J., Fraser, P. J., Langenfelds, R. L., and Sturges, W. T.: Chlorine isotope composition in chlorofluorocarbons CFC-11, CFC-12 and CFC-113 in firn, stratospheric and tropospheric air, Atmos. Chem. Phys., 15, 6867-6877, 10.5194/acp-15-6867-2015, 2015.

Buizert, C., Martinerie, P., Petrenko, V., Severinghaus, J., Trudinger, C., Witrant, E., Rosen, J., Orsi, A., Rubino, M., Etheridge, D., et al.: Gas transport in firn: multiple-tracer characterisation and model intercomparison for NEEM, Northern Greenland, Atmospheric Chemistry and Physics, 12, 4259–4277, 2012.

Eiler, J., Cesar, J., Chimiak, L., Dallas, B., Grice, K., Griep-Raming, J., Juchelka, D., Kitchen, N., Lloyd, M., Makarov, A., Robins, R., and Schwieters, J.: Analysis of molecular isotopic structures at high precision and accuracy by Orbitrap mass spectrometry, Int. J. Mass Spectrom., 422, 126-142, 10.1016/j.ijms.2017.10.002, 2017.

Hauri, E. H., Wang, J., Pearson, D. G., and Bulanova, G. P.: Microanalysis of δ13C, δ15N, and N abundances in diamonds by secondary ion mass spectrometry, Chem. Geol., 185, 149-163, 10.1016/S0009-2541(01)00400-4, 2002.

Horst, A., Lacrampe-Couloume, G., and Sherwood Lollar, B.: Compound-specific stable carbon isotope analysis of chlorofluorocarbons in groundwater, Anal. Chem., 87, 10498-10504, 10.1021/acs.analchem.5b02701, 2015.

Kaiser, J., Engel, A., Borchers, R., and Röckmann, T.: Probing stratospheric transport and chemistry with new balloon and aircraft observations of the meridional and vertical N2O isotope distribution, Atmospheric Chemistry and Physics, 6, 3535–3556, 2006.

Kaiser, J.: Reformulated 17O correction of mass spectrometric stable isotope measurements in carbon dioxide and a critical appraisal of historic 'absolute' carbon and oxygen isotope ratios, Geochimica et Cosmochimica Acta, 72, 1312-1334, 2008

Kirk-Othmer, Kirk-Othmer Encyclopedia of Chemical Technology. 4th ed. Volumes 1: New York, NY. John Wiley and Sons, 1991-Present., 11, 507, 1994

Nier, A. O.: A mass spectrometer for routine isotope abundance measurements, Rev. Sci. Instrum., 11, 212-216, 10.1063/1.1751688, 1940.

Phillips, E., Gilevska, T., Horst, A., Manna, J., Seger, E., Lutz, E. J., Norcross, S., Morgan, S. A., West, K. A., Mack, E. E., et al.: Transformation of Chlorofluorocarbons Investigated via Stable Carbon Compound-Specific Isotope Analysis, Environmental Science & Technology, 54, 870–878, 2019.

Schutten, J., Boerboom, A. J. H., v. d. Hauw, T., and Monterie, F.: Precise measurement of isotope ratios with a single collector mass spectrometer, Applied Scientific Research, Section B, 6, 388-392, 10.1007/BF02920395, 1957.

Volk, C. M., Elkins, J. W., Fahey, D. W., Dutton, G. S., Gilligan, J. M., Loewenstein, M., Podolske, J. R., Chan, K. R., and Gunson, M. R.: Evaluation of source gas lifetimes from stratospheric observations, Journal of Geophysical Research: Atmospheres, 102, 25543-25564, 1997

Witrant, E., Hogan, C., Laube, J., Kawamura, K., Capron, E., Montzka, S., Dlugokencky, E., Etheridge, D., Blunier, T., Sturges, W., et al.: A new multi-gas constrained model of trace gas non-homogeneous transport in firn: evaluation and behaviour at eleven polar sites, Atmospheric Chemistry and Physics, 12, 11 465–11 483, 2012.

Zuiderweg, A., Kaiser, J., Laube, J. C., Röckmann, T., and Holzinger, R.: Stable carbon isotope fractionation in the UV photolysis of CFC-11 and CFC-12, Atmos. Chem. Phys., 12, 4379-4385, 10.5194/acp-12-4379-2012, 2012.

Zuiderweg, A., Holzinger, R., Martinerie, P., Schneider, R., Kaiser, J., Witrant, E., Etheridge, D., Petrenko, V., Blunier, T., and Röckmann, T.: Extreme 13C depletion of CCl2F2 in firn air samples from NEEM, Greenland, Atmos. Chem. Phys., 13, 599-609, 10.5194/acp13-599-2013, 2013.

---

## Author Response (AR2)

**Response to reviewer**

We thank the reviewer for the additional comments on our manuscript. The concerns presented by Reviewer 1 in their second review are similar to those presented in their first. We provide a brief rebuttal to these points, but would also like to draw attention to our detailed responses during and at the end of the discussion phase (https://acp.copernicus.org/preprints/acp-2020-843/acp-2020-843-SC1- supplement.pdf and https://acp.copernicus.org/preprints/acp-2020-843/acp-2020-843-AC1-supplement.pdf).

In response to point 1) from the reviewer and following the suggestions of the editor, we have made several additions to the manuscript to clarify that our method only differs slightly from the method used in previous peer-reviewed publications of Allin et al. (2015) and Laube et al. (2010), which contain more details on the method. These changes are detailed in the track changes document. We feel that the method description given in the manuscript is sufficient, and in accordance with citing previously published work in scientific papers.

In response to point 2) from the reviewer we reference the comparison of our method against measurements of replicate samples made on a GC-IRMS system (Zuiderweg et al. 2011, 2012), now presented in Appendix B. This method comparison is consistent over a 50 ‰ range, which makes it highly unlikely that any scale correction is relevant for our results, given the measurement precision (as we already argued before).

Regarding the reviewer's concerns about our use of a Rayleigh fractionation model to characterise the apparent isotopic fractionation in the stratosphere, we note that that this is an empirical framework that has been successfully applied to other stratospheric gases with different lifetimes, including $CH_4$, $N_2O$ and $H_2$, as well as chlorine isotopologues of CFCs (see references in our previous comments). Isotopic fractionations of different processes are linearly additive (Kaiser et al. 2006). The $O(^1D)$ sink is small, between 2 % and 6 % of the total sink for CFC-11, -12, and -113 (Burkholder et al. 2013). Kaiser et al. (2006) also discuss in detail the effects of mixing and transport on the apparent isotope fractionations for $N_2O$ (which we know with much better precision). While mixing and transport are certainly relevant and reduce the observed apparent stratospheric isotope fraction compared with the intrinsic photochemical isotope effects, the *variations* in these mixing and transport effects are negligible for the precision that we report for CFCs, as can be inferred from the more precise observations for $N_2O$. We have added some text to the discussion about the effects of mixing and transport.

We are convinced that the data we present and their statistical interpretation are robust. While a higher measurement precision may be achievable with a larger sample size, our results are more than sufficient to draw the conclusions made in the manuscript.

**Response to editor**
We thank the editor for their helpful suggestions. We have implemented all of these in the revised manuscript.

Specifically, we have clarified at the end of the introduction and in the methods section that 1) our method differs only in minor ways from previous published methods (Allin et al. 2015, Laube et al. 2010); 2) the same method has been applied successfully and published previously; and 3) this method gives isotope delta values for a set of laboratory photolysis samples compatible with independent GC-IRMS analyses (as described in Appendix B).

Furthermore, we added examples of previous uses of single detector isotope mass spectrometry to Appendix B.

We have also changed the conclusions following the editor's suggestion.

The changes to our manuscript are highlighted in the track changes document submitted alongside this response. We trust to have addressed all of the editor's comments.

References

Allin, S., Laube, J., Witrant, E., Kaiser, J., McKenna, E., Dennis, P., Mulvaney, R., Capron, E., Martinerie, P., Roeckmann, T., et al.: Chlorine isotope composition in chlorofluorocarbons CFC-11, CFC-12 and CFC-113 in firn, stratospheric and tropospheric air, Atmospheric chemistry and physics, 15, 6867–6877, 2015.

Laube, J., Kaiser, J., Sturges, W., Bönisch, H., and Engel, A.: Chlorine isotope fractionation in the stratosphere, Science, 329, 1167–1167, 2010

Burkholder, J. B., Mellouki, W., Fleming, E. L., George, C., Heard, D. E., Jackman, C. H., Kurylo, M. J., Orkin, V. L., Swartz, W. H., and 25 Wallington, T. J.: Chapter 3: Evaluation of Atmospheric Loss Processes, SPARC Report on the Lifetimes of Stratospheric Ozone-Depleting Substances, Their Replacements, and Related Species, SPARC Report No. 6, 2013.

Kaiser, J., Engel, A., Borchers, R., and Röckmann, T.: Probing stratospheric transport and chemistry with new balloon and aircraft observations of the meridional and vertical N2O isotope distribution, Atmospheric Chemistry and Physics, 6, 3535–3556, 2006

Zuiderweg, A., Holzinger, R., and Röckmann, T.: Analytical system for carbon stable isotope measurements of light non-methane hydrocarbons, Atmospheric Measurement Techniques Discussions, 4, 101–133, 2011.

Zuiderweg, A., Kaiser, J., Laube, J., Röckmann, T., and Holzinger, R.: Stable carbon isotope fractionation in the UV photolysis of CFC-11 and CFC-12, Atmospheric Chemistry and Physics, 12, 4379–4385, 2012.